**communications**

**earth & environment**

# A 350,000-year history of groundwater recharge in the southern Great Basin, USA

Tracie R. Jackson [1,5], Simon D. Steidle [2,5✉], Kathleen A. Wendt[3], Yuri Dublyansky [2], R. Lawrence Edwards[4] & Christoph Spötl [2]

Estimating groundwater recharge under various climate conditions is important for predicting future freshwater availability. This is especially true for the water-limited region of the southern Great Basin, USA. To investigate the response of groundwater recharge to different climate states, we calculate the paleo recharge to a groundwater basin in southern Nevada over the last 350,000 years. Our approach combines a groundwater model with paleo-water-table data from Devils Hole cave. The minimum water-table during peak interglacial conditions was more than 1.6 m below modern levels, representing a recharge decline of less than 17% from present-day conditions. During peak glacial conditions, the water-table elevation was at least 9.5 m above modern levels, representing a recharge increase of more than 233–244% compared to present-day conditions. The elevation of the Devils Hole water-table is 3–4 times more sensitive to groundwater recharge during dry interglacial periods, compared to wet glacial periods. This study can serve as a benchmark for understanding long-term effects of past and future climate change on groundwater resources.

[1] Nevada Water Science Center, U.S. Geological Survey, 500 Date Street, Boulder City, NV 89005, USA. [2] Institute of Geology, University of Innsbruck, Innrain 52, 6020 Innsbruck, Austria. [3] College of Earth, Ocean, and Atmospheric Sciences, Oregon State University, 101 SW 26th Street, Corvallis, OR 97330, USA. [4] School of Earth and Environmental Sciences, University of Minnesota, 116 Church Street SE, Minneapolis, MN 55455-0149, USA. [5] These authors contributed equally: Tracie R. Jackson, Simon D. Steidle. ✉email: simon.steidle@student.uibk.ac.at

In arid regions, such as the southern Great Basin, USA, surface water is scarce, and groundwater is critical for maintaining ecosystem health and supporting socio-economic needs. Increasing groundwater demand to sustain population growth is exacerbated by climate change[1]. As an example, socio-economic demands combined with decreasing precipitation and increasing air temperatures have resulted in a megadrought that has persisted in the southern Great Basin since 2000[2]. Groundwater basins in this region are large and geologically complex, which presents an additional challenge for understanding climate-change effects. This is because large groundwater basins have long (>1000-year) equilibration timescales between time-varying recharge and groundwater-level changes[3]. The Ash Meadows groundwater basin (AMGB) in southwest Nevada (37°N, 116°W) is a prime example.

The AMGB encompasses desert valleys and highland areas in the north-central part of the Mojave Desert. The major discharge area in the AMGB is the Ash Meadows discharge area, which is a desert oasis supplied by high-discharge springs (Fig. 1) that support several endemic and endangered species, including the Devils Hole pupfish (*Cyprinodon diabolis*). A major threat to these species is a declining water table due, in part, to future climate change. Average temperatures in the western USA are projected to increase between 3.3 and 8.0 °C by 2100 in the CMIP6/AR6 SSP5-8.5 scenario[4], leading to severe and long-term droughts[5]. Warmer temperatures during the next century will contribute to a decrease in snowfall in highland areas of the western USA[4]. Because snowmelt from highland areas is the principal source of recharge to the AMGB[6,7] (Fig. 1), recharge also is expected to decrease[8].

Due to the large scale of the AMGB, understanding the dynamic link between groundwater levels and recharge volume requires a long-term (>1000-year) perspective. The climate history of the southern Great Basin is characterized by the repeated expansion and desiccation of large pluvial lakes throughout the Quaternary[9]. Climate changes are reflected in paleo-water-table elevations, which have been dated by uranium-series dating of subaqueous calcite deposits in Devils Hole[10] and Devils Hole 2[11] caves located in the Ash Meadows discharge area (Fig. 2a).

The Devils Hole paleo-water-table record[11] currently spans three glacial-interglacial cycles (Fig. 2a). During peak dry (interglacial) periods, the paleo water table in Devils Hole 2 was

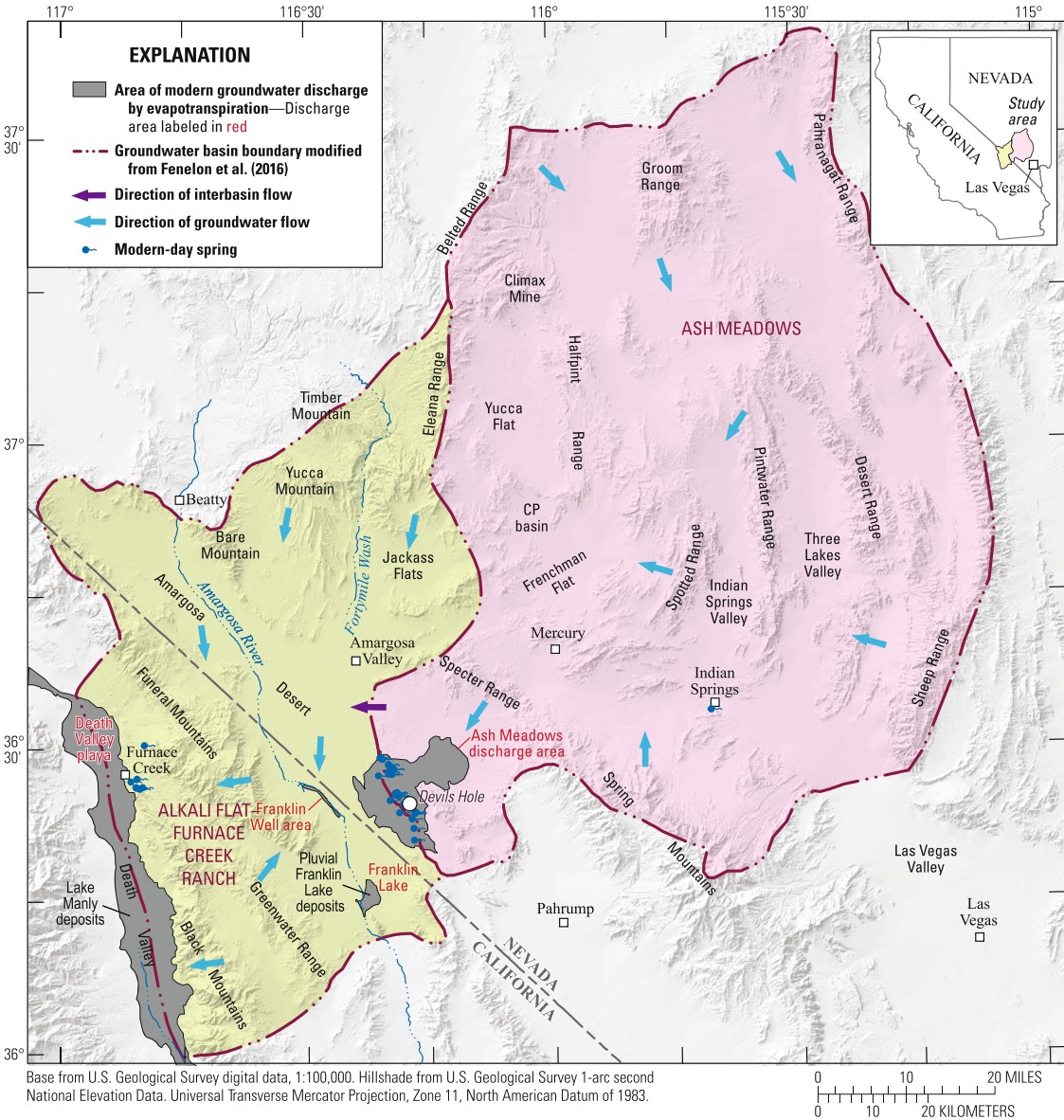

**Fig. 1 Map of the study area.** Map of the study area, which includes the Ash Meadows and Alkali Flat-Furnace Creek Ranch groundwater basins in southern Nevada and California, USA. Figure modified from Halford and Jackson[7].

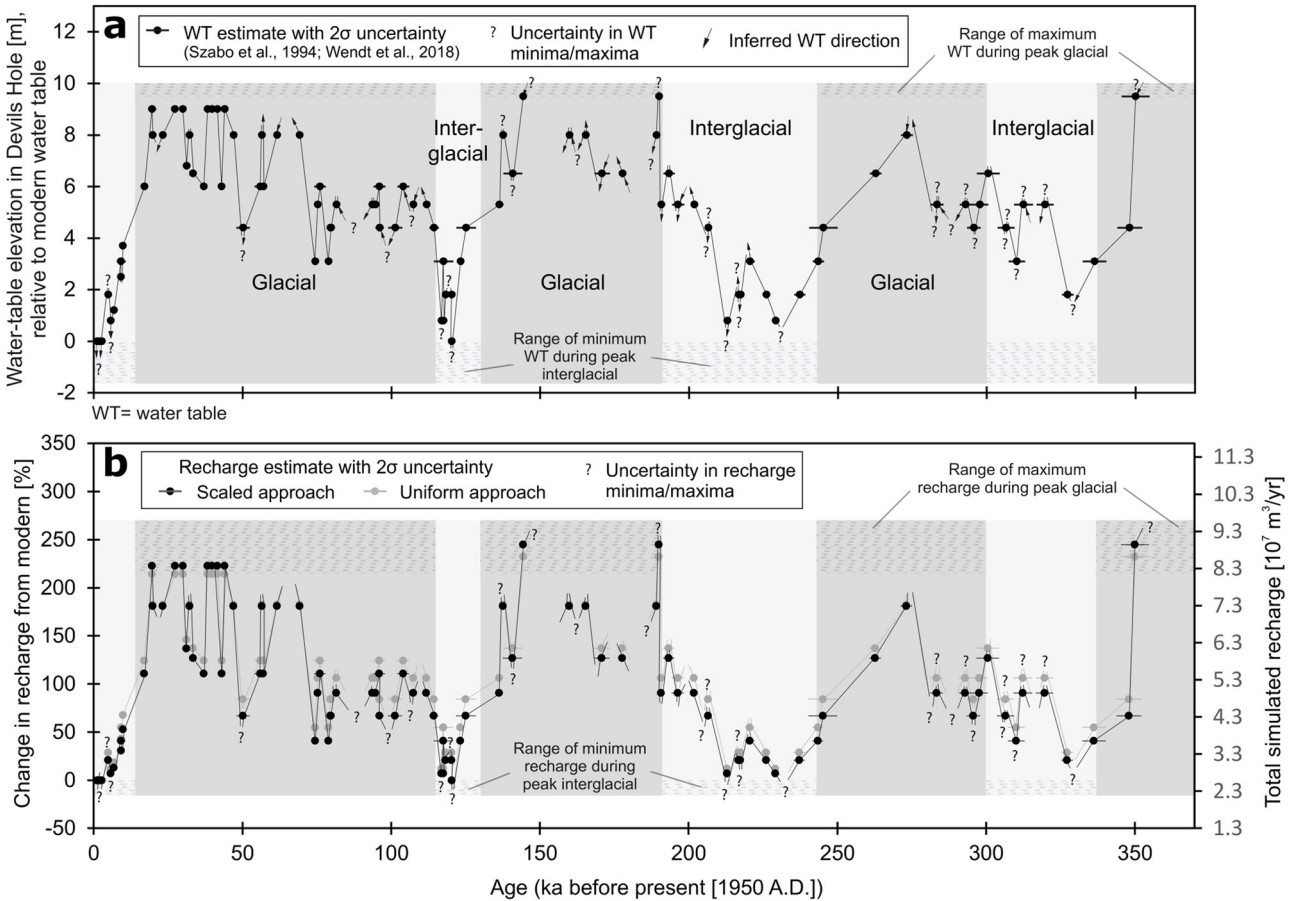

**Fig. 2 History of paleo-water table and recharge volume.** Devils Hole paleo-water-table elevations, relative to the modern level at 0 m (**a**). Total and percent change in Ash Meadows groundwater basin recharge from modern-recharge volume (0% corresponds to $2.59 \times 10^7$ m³/yr) (**b**). Computed recharge estimates from relation developed in Fig. 3 and paleo-water-table data in (**a**).

below the modern level of 0 m but above −1.6 m. During wet (glacial) periods, the paleo water table was above +9.5 m, although the exact maximum limit is unknown.

Previous attempts to constrain paleo-recharge estimates in the southern Great Basin have resulted in a large range of estimates[8,12–16] and were biased toward glacial periods. Paleorecharge estimates during past interglacials have remained largely unstudied. At Devils Hole, Wendt et al.[17] qualitatively related the Devils Hole paleo-water-table record to recharge, but the study is not suitable for quantifying basin-scale recharge volumes. This study uses a novel approach that modifies a groundwater-flow model[7] to estimate AMGB volumetric recharge rates that correspond to Devils Hole paleo-water-table elevations during the last 350,000 years[11]. These volumetric paleo-recharge rates are the first-ever quantitative recharge estimates in the southern Great Basin over the last 350,000 years. A relation is developed between Devils Hole paleo-water-table elevations and recharge that can be used to forecast Devils Hole water-table changes in response to future changes in recharge. Even though this study is focused on the southern Great Basin, the novel approach used shows potential for translating paleo-water-table data into quantifiable paleo-recharge estimates, and these estimates can be used as a benchmark for understanding the effects of future climate change on groundwater systems.

## Results

**Relation between paleo-water-table change and recharge.** Recharge (m³/yr) in the AMGB was estimated for Devils Hole

water-table changes of between −2 and +10.7 m r.m.w.t. (relative to the modern-day water table). Scaled-recharge and uniform-recharge approaches were used to derive relations between Devils Hole water-table change and recharge (Fig. 3). A volumetric modern-recharge rate of $2.59 \times 10^7$ m³/yr[7] is the reference recharge (0%) and corresponds to the modern level (0 m r.m.w.t.) for the purpose of estimating percent change in recharge relative to modern. In this study, water-table minima for peak glacial and interglacial conditions are defined at +9.5 and −1.6 m r.m.w.t., respectively.

During peak glacial conditions, water-table elevations were at least +9.5 m r.m.w.t. (Fig. 2a). A lower limit of +9.5 m r.m.w.t. relates to ≥244% and ≥233% increases in scaled- and uniform-recharge, respectively, from modern conditions (Fig. 3). Peak interglacial conditions are associated with paleo-water-table elevations ranging from 0 to −1.6 m r.m.w.t. (Fig. 2a). A lower limit of −1.6 m r.m.w.t. relates to <17% and <22% decreases in scaled and uniform recharge, respectively, from modern conditions (Fig. 3).

The relation of Devils Hole water-table change to AMGB recharge is nonlinear (Fig. 3). Assuming maximum water-table elevations fluctuated between +9 and +10 m r.m.w.t. during glacial periods, the scaled-recharge relation has a slope of 2.2 cm of water-table change per percent change in recharge (2.2 cm/%) and the uniform-recharge relation has a slope of 2.7 cm/%. Assuming minimum water-table elevations fluctuated between −1.6 and 0 m r.m.w.t. during interglacial periods, the scaled- and uniform-recharge relations have slopes of 9.3 cm/% and 7.3 cm/%, respectively. Comparison between peak glacial-period relations and peak interglacial-period relations indicates the numerical

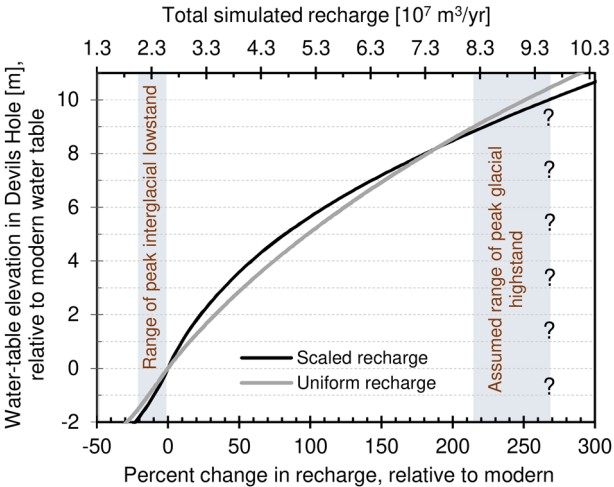

**Fig. 3 Total simulated recharge for different paleo-water-table elevations.** Total simulated recharge to Ash Meadows groundwater basin and percent change from modern (0%) recharge for different paleo-water-table elevations at Devils Hole relative to the modern water table at 0 m.

model is more sensitive to the simulation of spatially distributed recharge during drier periods.

The relation of Devils Hole water-table change to AMGB recharge (Fig. 3) was used to estimate paleo recharges corresponding to Devils Hole paleo-water-table elevations during the last 350,000 years (Fig. 2b). As expected, more paleo recharge is estimated during wet glacial periods and less paleo recharge is estimated during dry interglacial periods (Fig. 2b).

Groundwater-discharge areas were simulated and compared between modern-day and a glacial scenario (+9.5 m r.m.w.t.) using the scaled-recharge and uniform-recharge approaches (Fig. 4). Additional groundwater-discharge areas not existing today were simulated in the glacial scenarios. These discharge areas are in the Amargosa Desert and south of Indian Springs and Three Lakes Valleys (Fig. 4b, c). Simulated groundwater-discharge areas have a larger footprint in the uniform-recharge scenario (Fig. 4c) compared to the scaled-recharge scenario (Fig. 4b), because a higher ratio of recharge is simulated to occur directly on valley floors (see Supplementary Figs. 7 and 9). Both scaled-recharge and uniform-recharge approaches appear to provide reasonable results, based on a comparison of simulated and mapped paleodischarge areas.

Groundwater-discharge areas also were simulated and compared between modern day and an interglacial scenario (−1.6 m r.m.w.t.) using the scaled-recharge and uniform-recharge approaches (Fig. 5). All modern discharge areas (Fig. 5a) were simulated in the interglacial scenarios (Fig. 5b, c), except that discharge rates were less than modern discharge rates. The scaled-recharge scenario follows expectations because no additional discharge areas were simulated outside of modern discharge areas (Fig. 5a, b). The uniform-recharge scenario simulates additional discharge areas in Indian Springs and Three Lakes Valleys (Fig. 5c) that likely are implausible. If these simulated discharge areas occurred during a drier interglacial period, then the discharge areas would be present during modern, wetter climatic conditions. Results indicate that a scaled-recharge approach better simulates drier periods, compared to the uniform-recharge approach.

## Discussion

**Model assumptions.** This work assumes that changes in rock transmissivity over the past 350,000 years were minimal and did

not significantly affect the Devils Hole paleo-water-table record. This work also assumes that the water-table record was not significantly affected by tectonic deformation that caused local changes to the land-surface altitude or volumetric strain in the aquifer. Carbonate-rock dissolution or fracturing and faulting due to tectonic activity could alter transmissivity[18]. Incremental changes in transmissivity, land-surface adjustments, or volumetric strain over the past 350,000 years are expected to impose a long-term downward or upward trend in the Devils Hole water-level record. For example, Robertson et al.[19] estimated that the rate of water-level decline in Devils Hole due to volumetric strain could be as high as 0.02 cm/yr. If this rate were sustained for 350,000 years, the water level in Devils Hole would have declined 70 m. A water-level change of this magnitude is not observed (Fig. 2a). The water-level record is dominated by large oscillations that correspond to wet and dry climatic conditions, whereas the long-term trend is relatively flat. Any potential long-term rise or decline that is masked by these large oscillations would have to be small (<2 m over 350,000 years). Therefore, the assumption is reasonable that changes to transmissivity, land-surface altitude, and volumetric strain over the past 350,000 years were minimal relative to changes in the Devils Hole water-table record.

**Qualitative evaluation of simulated paleodischarge.** To evaluate the accuracy of simulated paleodischarge areas in the peak glacial (+9.5 m r.m.w.t.) scenarios (Fig. 4), simulated paleodischarge locations were qualitatively compared to locations of modern discharge and documented paleo-spring deposits. As indicated in Fig. 4b, c, some of the simulated discharge locations coincide with modern-day regional springs, whereas other simulated discharge locations are consistent with paleodischarge areas.

Deposits of aquatic and land snails in the study area document former springs. Such deposits were found and dated to glacial periods[20–23]. Combined, mapped snail deposits cover locations in the Amargosa Desert, and areas south of Indian Springs and Three Lakes Valleys. These paleodischarge locations coincide with locations of simulated discharge (Fig. 4b, c).

Carbonate deposits of tufa and calcite veins indicate locations of modern and paleo-groundwater discharge. Mapped carbonate deposits north of Ash Meadows discharge area indicate that the paleodischarge area extended farther north than the modern discharge area[18], which is consistent with simulated paleodischarge areas that extend farther north (Fig. 4). Paces et al.[24] and Paces and Whelan[25] describe a variety of groundwater-discharge deposits in the Amargosa Desert, where mapped paleodischarge locations generally are consistent with simulated paleodischarge areas (Fig. 4).

Travertine deposits in Mercury Valley[26] suggest former discharge that was not simulated in the peak glacial (+9.5 m r.m.w.t.) scenario (Fig. 4b, c). Most of these travertine deposits, however, are older than 700,000 years and may have been subject to geological processes, such as tectonic uplift, that are not relevant to this study. Overall, simulated groundwater discharge in the peak glacial scenarios generally are consistent with documented paleodischarge sites during the late Pleistocene (Fig. 4b, c).

**Model uncertainty.** The largest uncertainty associated with the modeling effort is the simulation of spatially distributed recharge. In modern settings, "recharge is the most difficult component of the groundwater system to quantify" (Bredehoeft[27], p. 1). To help constrain estimated recharge volumes, two different recharge conceptualizations were used: scaled- and uniform-recharge changes, relative to modern conditions. Both recharge conceptualizations assume that land-surface topography was stable

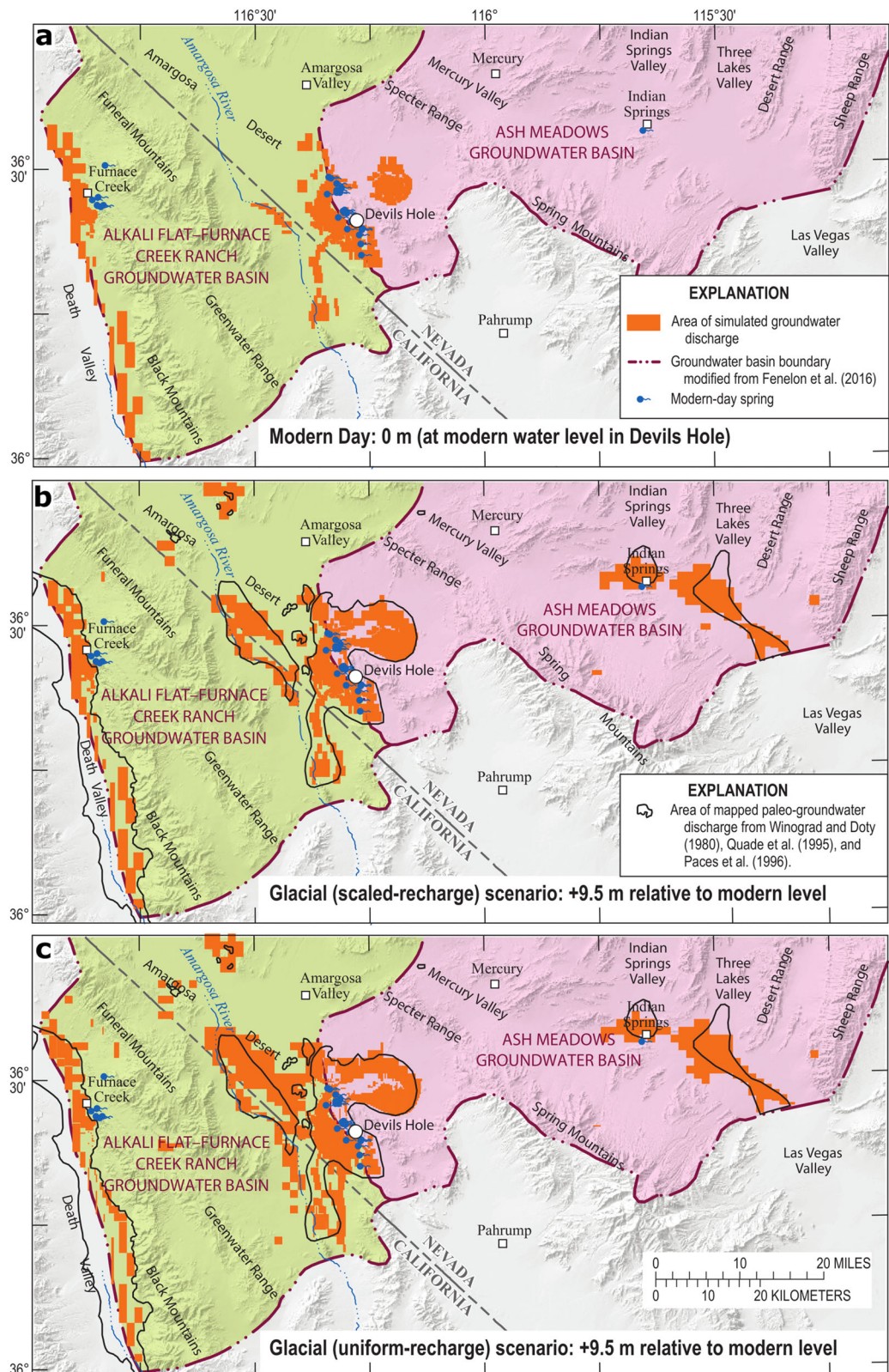

**Fig. 4 Groundwater-discharge areas for modern-day and glacial condition.** Simulated groundwater-discharge areas for modern-day (**a**), glacial (scaled-recharge; **b**), and glacial (uniform recharge; **c**) scenarios.

during the last 350,000 years because the last period of extension and uplift occurred from the Miocene to Pliocene, between 23 and 2.6 million years ago[28]. Saturated hydraulic properties also are relatively stable. If the topography is the same, then the recharge distribution is still controlled by high and low topographic altitudes, despite atmospheres that are either cooler and wetter or warmer and drier. However, the relative recharge contribution between valley floors and highland areas may have

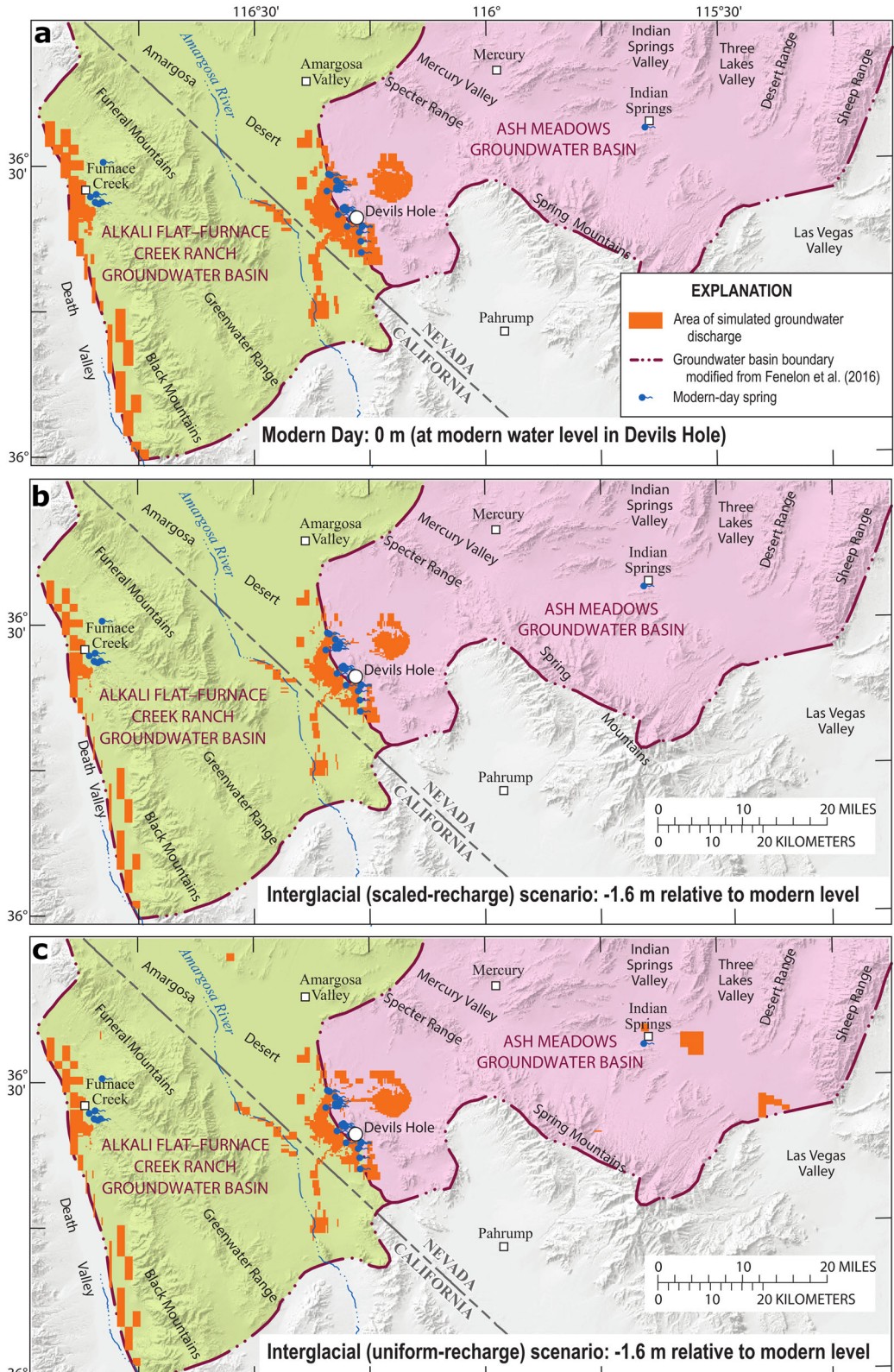

**Fig. 5 Groundwater-discharge areas for modern-day and interglacial condition.** Simulated groundwater-discharge areas for modern-day (**a**), interglacial (scaled-recharge; **b**), and interglacial (uniform recharge; **c**) scenarios.

changed. The uniform-recharge approach assumes that valley floors receive a larger recharge contribution in wetter periods. Conversely, the scaled-recharge approach assumes that the ratio of the spatial recharge distribution has remained unchanged

during the last 350,000 years (i.e., recharge on valley floors is small-to-negligible compared to highland areas).

A qualitative evaluation of simulated paleodischarge areas in the peak glacial (+9.5 m r.m.w.t.) scenarios (Fig. 4b, c) indicates

that both scaled- and uniform-recharge distributions can reasonably simulate wetter climatic periods. This result is consistent with relatively similar relations between Devils Hole water-level changes and recharge during glacial conditions (2.2–2.7 cm/%).

A qualitative evaluation of simulated paleodischarge areas in the peak interglacial (−1.6 m r.m.w.t.) scenarios (Fig. 5b, c) indicates that a scaled-recharge approach better simulates drier periods, compared to the uniform-recharge approach. The scaled-recharge scenario follows expectations of simulating paleo-discharge areas only within the spatial extent of modern discharge areas (Fig. 5a, b). The uniform-recharge scenario implausibly simulates additional paleodischarge areas in Indian Springs and Three Lakes Valleys (Fig. 5c), which are not present in a wetter, modern climate (Fig. 5a). Therefore, a relation of 9.3 cm/% using the scaled-recharge approach likely better represents the correlation between Devils Hole water-level changes and recharge during interglacial periods.

**Comparison with paleo-recharge studies.** Yucca Mountain (50 km northwest of Devils Hole; Fig. 1) is one of the few areas in the southern Great Basin where paleo recharge has been estimated. Available studies focus on the last glacial period, when peak water-table elevations at Devils Hole likely fluctuated between +9 and +10 m r.m.w.t.[10,11] Water-table elevations from +9 to +10 m r.m.w.t. relate to increases in recharge of +215–270%, relative to modern (Fig. 3).

Glacial-period recharge estimates from this study (+215–270%) are compared to previous studies of Yucca Mountain. Glacial-period precipitation is estimated to be +100–160% greater, compared to modern day, based on interpretation of plant macrofossils near Yucca Mountain[29,30]. Approximating recharge as precipitation minus evapotranspiration (P-ET), a +100–160% increase in precipitation translates to a greater increase in recharge, because evaporation is expected to be lower in a cooler climate. Czarnecki[31] and Schwartz et al.[13] estimated glacial-period recharge rates in Yucca Mountain to be an order of magnitude greater than modern rates using regional groundwater models. The order-of-magnitude estimate is biased high because these studies developed regional groundwater models using basin fluxes and groundwater budgets that are biased high based on recent revisions[7,32], but were the best estimates at the time these studies were published. Paces et al.[33] collected secondary opal deposits from the deep unsaturated zone of Yucca Mountain, which were later analyzed by Maher et al.[15]. Maher et al.[15] estimated net infiltration rates during the winter season that were up to 100% greater than modern day; however, these net infiltration rates were based on samples that are "snapshots in time" and do not necessarily coincide with peak glaciation (i.e., the maximum Devils Hole water-table elevation).

Oster et al.[14] used multiple climate models that span the western USA to compare modern and glacial-period precipitation patterns. The five models that show the best agreement with proxy records are CNRM-CM5, IPSL-CM5A-LR, MPI-ESM-P, NCAR CCSM4, and MIROC-ESM. The P–ET in four of these five models range from +200 to +300% (ref. 14, Supplementary Fig. S2), which is consistent with the +215–270% increase in recharge estimated in this study (Fig. 2b). The MIROC-ESM climate model simulates an exceedingly high P–ET increase of more than +750%.

**Relation between Devils Hole water-table change and recharge.** The nonlinear relation of Devils Hole water-table change to AMGB recharge is about three-to-four times as sensitive to recharge changes during drier, interglacial periods (9.3 cm/%), compared to wet, glacial periods (2.2–2.7 cm/%). This nonlinearity occurs because Devils Hole is within the Ash Meadows discharge area, which is the primary discharge area in AMGB (Fig. 1). Water-table changes below modern conditions have been relatively small (−1. to 0 m) because discharges from nearby Ash Meadows springs have upheld the water table within the last 350,000 years. The rate of water-table change above modern conditions steepens with increasing recharge because more water is needed to elevate the water table over a large spatial area since wetter conditions form more discharge areas that stabilize the water table.

## Conclusions

Understanding climate-change effects on groundwater levels is critical in the southwest USA, where recharge is expected to decrease substantially during the next several centuries. This study uses a novel approach of coupling published paleo-water-table data with a modified groundwater model to provide the first quantitative recharge estimates in the southern Great Basin over the last 350,000 years. Previous attempts to quantify paleo-recharge in this region were limited to indirect or intermittently available proxies that are biased toward past pluvial periods. This study spans the last three glacial-to-interglacial cycles and provides paleo-recharge estimates for a large range of climate states. These paleo-recharge estimates are based on reasonable assumptions that transmissivity, land-surface altitude, and volumetric strain over the past 350,000 years were minimal, relative to changes in the Devils Hole water-table record.

Paleo recharge was estimated using scaled- and uniform-recharge approaches to account for uncertainties associated with spatially distributed recharge, compared to modern conditions. Both recharge conceptualizations assume that the recharge distribution is controlled by high and low topographic altitudes, despite atmospheres that are either cooler and wetter or warmer and drier. However, the uniform-recharge approach assumes that valley floors receive a larger recharge contribution in wetter periods than the scaled-recharge approach, which assumes that spatially distributed recharge has remained small-to-negligible during the last 350,000 years. Both recharge assumptions were tested and validated by comparing model-simulated and mapped paleodischarge areas during glacial and interglacial periods. General agreement between simulated and mapped paleo-discharge areas indicates that scaled- and uniform-recharge distributions can reasonably simulate wetter climatic periods. The scaled-recharge approach better simulates drier periods, compared to the uniform-recharge approach, because uniform recharge implausibly simulates additional paleodischarge areas outside of modern discharge areas.

Paleorecharge to the AMGB was estimated for the Devils Hole paleo-water-table record that spans the last 350,000 years. The lower limit of the paleo water table was most likely above −1.6 m r.m.w.t. during peak interglacial conditions, which relates to a decrease in estimated recharge of no more than 17%, relative to modern, using the scaled-recharge assumption. A water table lower than −1.6 m r.m.w.t. cannot be ruled out but is considered unlikely. The paleo water table was at least +9.5 m r.m.w.t. during peak glacial conditions, which relates to an increase in recharge of at least 233–244%, relative to modern, using results from scaled- and uniform-recharge approaches.

Relations were derived between Devils Hole water-table change and AMGB recharge. The relations are expressed as cm of Devils Hole water-table change per percent change in AMGB recharge (cm/%). Similar relations between Devils Hole water-table change and AMGB recharge during glacial conditions (2.2 and 2.7 cm/%) indicate the relative insensitivity of how recharge is spatially distributed during wetter climates. A relation of 9.3 cm/%

estimated from the scaled-recharge approach better represents the correlation between Devils Hole water-level changes and AMGB recharge during interglacial periods. Comparison of the peak glacial and interglacial slopes indicates that water-table changes are three-to-four times as sensitive to recharge changes during drier, interglacial periods (−1.6 to 0 m r.m.w.t.), compared to recharge changes during wet, glacial periods (+9 to +10 m r.m.w.t.). These slopes can be used to forecast future climate effects on water-table changes in Devils Hole, using recharge estimates from climate models. Paleorecharge estimates from this study also can be used as a benchmark for understanding effects of future climate change on groundwater systems outside the southern Great Basin.

## Methods

**Study area description and justification.** The study area (Fig. 1) is focused on the AMGB but includes the Alkali Flat-Furnace Creek Ranch groundwater basin (AFFCRGB), because these basins are hydraulically connected[7]. Altitudes in the AMGB range from 660 m in the Ash Meadows discharge area to about 3425 m in the Spring Mountains. Altitudes in the AFFCRGB range from −86 m on the Death Valley floor to 2315 m in the Eleana Range.

Expanding the study area to include adjacent bounding basins would have no effect on the water levels in Devils Hole. For example, south of Devils Hole, interbasin flow from Pahrump Valley is negligible under modern conditions because geologic and hydrologic evidence indicate that a low-permeability hydraulic barrier precludes groundwater movement northward from Pahrump Valley to Devils Hole[32,34] (Fig. 1). Therefore, interbasin flow does not occur near Devils Hole. Devils Hole water levels also will be unaffected by potential, but unlikely, interbasin flows into the study area from basins distant from Devils Hole, such as Railroad Valley[7,32,35]. During wetter periods, groundwater-flow paths shorten between recharge and discharge areas, and more discharge areas form. Therefore, the contributing area to Devils Hole will be smaller. During drier periods, significant interbasin flows are unlikely across basin boundaries because modern discharge areas will either decrease in areal extent, remain the same size but have decreased discharge rates, or disappear due to lowered water levels. Therefore, inclusion of the AMGB and AFFCRGB is adequate for estimating recharge volumes under different climate scenarios. See 'Supplementary Note' for a more detailed explanation that justifies the study area extent.

Prior to groundwater development in 1950, groundwater conditions in AMGB and AFFCRGB were approximately at steady state, where recharge rates were balanced by discharge rates and interbasin flows[7]. The AMGB (11,500 km²) has an estimated modern-day recharge of $2.59 \times 10^7$ m³/yr, where $3.15 \times 10^6$ m³/yr of interbasin flow moves from the AMGB into the AFFCRGB and $2.28 \times 10^7$ m³/yr discharges from springs and evapotranspiration areas in the Ash Meadows discharge area[7]. About 80% of AMGB recharge occurs in the Spring Mountains and Sheep Range and is sourced primarily from snowmelt[6,7]. Additional background information on the study area is provided in the "Supplementary Note".

**Limitations of the paleo-water-table record.** The 350,000-year Devils Hole water-table record was constructed by Szabo et al.[10] and Wendt et al.[11] using calcite cores that were sampled from cave walls at discrete elevations (Fig. 2a). Due to the nature of discrete sampling, the compiled record does not capture the precise elevation of paleo-water-table fluctuations. For example, the two lowest cores were sampled at +0 m and −1.6 m relative to the modern water table. The lowest core does not contain folia, nor are folia observed on the exposed cave walls below −1.6 m. Therefore, the paleo-water-table is assumed to have never dropped below −1.6 m. Yet it is possible that the paleo water table fluctuated between modern-day levels and −1.6 m during periods of greater aridity. For example, evidence suggests that the Great Basin was drier during the Middle Holocene relative to today[36]. This period is poorly resolved in the Devils Hole paleo-water-table record (~3kyr sampling resolution). Any potential folia deposits between +0 m and −1.6 m during the Middle Holocene cannot be discerned in the current record. Furthermore, rapid fluctuations of the paleo-water-table could result in microscopically thin or potential absence of folia deposits at sampled elevations[11]. We acknowledge this lack of complete spatial and temporal continuity, and instead use the Devils Hole paleo-water-table record to investigate recharge conditions under a range of climate states, rather than individual events, during the last 350,000 years.

**Defining the modern water table.** This study estimates AMGB recharge based on paleo-water-table changes from the modern level in Devils Hole. Groundwater pumping is not considered in this study; thus, the predevelopment (pre-1950) setting only is involved. The paleo-water-table reconstruction is based on calcite that precipitated on the walls of Devils Hole[10] and Devils Hole 2 cave[11]. These subvertical tectonic caves are ~200 m apart and developed along a set of northeast-striking normal faults. Given the close proximity and hydraulic connectivity of Devils Hole 2 cave and Devils Hole, water-table altitudes at both locations are the same and the location is hereafter only referred to as Devils Hole (36.416°N, 116.283°W). The modern water table of Devils Hole is defined at the 1937 altitude of 719 m[37].

**Model description.** The Death Valley version 3 steady-state (DV3-SS), groundwater-flow model[7] was used to estimate recharge during past glacial and interglacial periods. The DV3-SS model was developed to simulate modern groundwater flow, where modern conditions are defined as occurring within the last 100 years but prior to development. Conditions during past glacial and interglacial periods can be simulated with minor modifications to the model by assuming that equilibration to steady-state conditions is achieved during each glacial and interglacial period. Steady-state assumption validity was tested by running the DV3-SS model as a transient model for different recharge conditions. The time for Devils Hole water levels to reach near equilibration to a steady-state condition is within 1000 years. See 'Supplementary Discussion' for more details.

Steady-state conditions assume that the groundwater system is in a state of dynamic equilibrium. Dynamic equilibrium recognizes that groundwater levels are not stationary, but fluctuate with time, because of short-term (decadal) and long-term (millennial-scale) changes in recharge and discharge. During each glacial and interglacial, the groundwater system equilibrates to the changing climate, where long-term cumulative recharge is balanced by long-term cumulative discharge and the net change in long-term cumulative groundwater storage is zero.

The DV3-SS model[7,38] is a 3D, finite-difference, groundwater-flow model[39] (MODFLOW-2005). The model simulates modern conditions in the Death Valley regional flow system and includes the AMGB and AFFCRGB (Fig. 1). The top surface of the model is the modern water table. The spatial distribution of recharge was simulated with the MODFLOW recharge package[39]. Discharges from springs and evapotranspiration areas were specified using the well package in MODFLOW[39] (Supplementary Fig. 4). Discharges from desert playas in Death Valley and Franklin Lake were simulated as specified heads because discharge estimates are uncertain[40], but water-table altitudes are known and are within a few meters of land surface.

Heterogeneous recharge-rate and hydraulic-conductivity distributions were estimated during calibration of the DV3-SS model. Measured water-level altitudes in wells, spring pools, and evapotranspiration areas, and transmissivity estimates from aquifer tests and specific capacity were compared to simulated equivalents during model calibration. DV3-SS model construction is fully described in the "Supplementary Methods".

**Model modifications.** The DV3-SS model was modified to simulate water-level changes from recharge during past glacial and interglacial periods[41]. The modified model was developed by converting the original DV3-SS model into a super-position model. The model has initial heads of 0 m, which are conceptualized as no water-level change from modern conditions. Simulated water-level changes are relative to modern heads, where the modern (predevelopment) head distribution was obtained from the calibrated DV3-SS model[7].

The aquifer stress in the superposition model is recharge. Recharge distributions were simulated during past glacial and interglacial periods using two approaches: scaled recharge and uniform recharge. These two approaches were used to account for uncertainty associated with changes in spatially distributed recharge.

In the scaled-recharge approach, a multiplication factor in the MODFLOW recharge package was changed manually by increments of 2% from −24% to +300% of modern recharge. These percent changes were used to scale the calibrated, modern-recharge distribution. Scaling recharge from −24 to +300% resulted in simulated water-table changes in Devils Hole that ranged from 2 m below to 10.7 m above the modern level. Simulated water-level changes in Devils Hole are relative to a modern water-table altitude of 719 m. In summary, the calibrated, modern-recharge distribution was multiplied by a scaling factor, then the model was run, the simulated head at Devils Hole was extracted, and the process repeated to obtain a relation between recharge and Devils Hole water levels.

In the uniform-recharge approach, a uniform amount of recharge was added to or subtracted from the calibrated, modern-recharge distribution. A series of recharge arrays were created by adding a uniform-recharge amount in increments of 0.0001 m from −0.0018 to 0.0066 m to the calibrated, modern-recharge distribution. These recharge increments simulate −29% to +280% of modern recharge, which correspond to simulated water-table changes in Devils Hole of 2 m below and 10.7 m above the modern level, respectively. For each model run, one recharge array was called by the MODFLOW recharge package, and the simulated head at Devils Hole was extracted. The process was repeated, with one model run for each recharge array, to obtain a relation between recharge and Devils Hole water levels.

Both scaled-recharge and uniform-recharge scenarios are based on a conceptual model where greater amounts of recharge occur in highland areas[7], but with different ratios of recharge between valley floors and highland areas. Scaling the calibrated, modern-recharge distribution to match the range of paleo-water-table changes in Devils Hole assumes that the spatial distribution of recharge has remained unchanged during the last 350,000 years. In this case, winter snowpack generates most of the recharge and a higher ratio of precipitation is converted to

recharge in highland areas. A lesser ratio of precipitation is converted into recharge as altitude decreases from highland areas to the valley floors because a progressively larger percentage of precipitation is lost to evapotranspiration. Adding uniform-recharge rates to the calibrated, modern-recharge distribution assumes the recharge contribution from the mountains is less significant relative to the valleys during wet periods. These recharge conceptualizations are consistent with how paleo recharge has been simulated in climate models in the western USA[14].

Boundary conditions from the DV3-SS model were changed in the superposition model to better simulate paleodischarge areas during past glacial periods. For example, Death Valley was covered by a large pluvial lake, Lake Manly, during cooler and wetter climate conditions of the Pleistocene[42] (Fig. 1). Death Valley and Franklin Lake were simulated as specified heads of 0 m (Supplementary Fig. 5), where lake formation is simulated as an increase in simulated discharge from the specified head boundaries. Simulation of lake features in the superposition model does not affect simulated water-table changes in Devils Hole because lake features are sufficiently far from Devils Hole.

Boundary conditions of modern springs and evapotranspiration areas were changed in the superposition model to account for changes in discharge during past glacial and interglacial periods. Discharge from springs and evapotranspiration areas were changed from specified-flow rates in the DV3-SS model to head-dependent boundaries in the superposition model (Supplementary Fig. 5). Head-dependent boundaries were simulated using the drain package in MODFLOW[39]. The drain package simulates groundwater discharge from the aquifer at a rate that is proportional to the difference between the simulated head in the aquifer and a user-defined specified drain elevation. Groundwater discharges from the aquifer if the simulated head is greater than the drain elevation, whereas no discharge occurs if the simulated head is lower than the drain elevation. The drain elevation was assigned equal to the modern water table at 0 m for modern springs and evapotranspiration areas.

To ensure all potential paleo-groundwater-discharge areas were simulated, a head-dependent boundary was specified with a drain for every cell across the entire top surface of the model domain that did not have another boundary condition within the same model cell (Supplementary Fig. 5). The specified drain elevation for these potential discharge areas was equal to the difference between land-surface altitude and the simulated, modern water-table altitude from the calibrated DV3-SS model. Essentially, the specified drain elevations are equal to depths to water, where groundwater discharge occurs at a model cell if the simulated recharge causes the simulated head to exceed land surface. More details regarding modifications to the DV3-SS model are described in the "Supplementary Methods" section. Model files are documented in a separate data release[41].

## Data availability

All data used in this paper have been published in previous work cited in the references. Data of this study can be found at https://doi.org/10.5066/P98YZC5P.

## Code availability

The code used to generate the presented data can be found at https://doi.org/10.5066/P98YZC5P.

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

## Acknowledgements
This research was funded by grant P327510 of the Austrian Science Fund. We thank the reviewers for their constructive comments, which have improved the quality of this manuscript. We also thank Joseph M. Fenelon, U.S. Geological Survey (retired), for drafting Fig. 2 of this paper.

## Author contributions
T.R.J. developed the numerical model, extracted model results, and wrote parts of the paper. S.D.S. ran model simulations, performed analyses, and wrote parts of the paper. K.A.W. conceptualized the study and wrote parts of the paper. Y.D., R.L.E., and C.S. provided technical expertise and contributed to the writing and reviewing of the paper.

## Competing interests
The authors declare no competing interests.
