## [Peer Review File · Communications Earth & Environment]

18th Mar 22

Dear Mr Steidle,

Your manuscript titled "A 350,000-year history of groundwater recharge in the southern Great Basin, USA" has now been seen by 3 reviewers, and I include their comments at the end of this message. They find your work of interest, but also raise important points which require revision. We are interested in the possibility of publishing your study in Communications Earth & Environment, but would like to consider your responses to these concerns and to our editorial thresholds, and assess the revised manuscript before we make a final decision on publication.

We therefore invite you to revise and resubmit your manuscript, along with a point-by-point response that takes into account the points raised. Please highlight all changes in the manuscript text file.

Please, also consider our editorial thresholds:

- provide compelling discussion of the apparent lack of response of the water-table level to mid-Holocene drying, including potential limitations of your material and consequent interpretation caveats
- present a demonstrably robust model of ground water recharge in the southern Great Basin which advances understanding beyond existing literature on the past and modern hydrological regime at Devil's Hole
- address the availability of updated climate models and assessment report scenarios (CMIP6/AR6)

Please use the following link to submit your revised manuscript, point-by-point response to the referees' comments (which should be in a separate document to any cover letter) and the completed checklist:

[link redacted]

We hope to receive your revised paper within six weeks; please let us know if you aren't able to submit it within this time so that we can discuss how best to proceed. If we don't hear from you, and the revision process takes significantly longer, we may close your file. In this event, we will still be happy to reconsider your paper at a later date, as long as nothing similar has been accepted for publication at Communications Earth & Environment or published elsewhere in the meantime.

We understand that due to the current global situation, the time required for revision may be longer than usual. We would appreciate it if you could keep us informed about an estimated

timescale for resubmission, to facilitate our planning. Of course, if you are unable to estimate, we are happy to accommodate necessary extensions nevertheless.

Please do not hesitate to contact me if you have any questions or would like to discuss these revisions further. We look forward to seeing the revised manuscript and thank you for the opportunity to review your work.

Best regards,

Ola Kwiecien, PhD
Editorial Board Member
Communications Earth & Environment
orcid.org/0000-0001-6018-9181

Joe Aslin
Senior Editor
Communications Earth & Environment

EDITORIAL POLICIES AND FORMATTING

Editorial Policy: [Policy requirements](https://www.nature.com/documents/nr-editorial-policy-checklist.zip)

Furthermore, please align your manuscript with our format requirements, which are summarized on the following checklist:

[Communications Earth & Environment formatting checklist](https://www.nature.com/documents/commsj-phys-style-formatting-checklist-article.pdf)

and also in our style and formatting guide [Communications Earth & Environment formatting guide](https://www.nature.com/documents/commsj-phys-style-formatting-guide-accept.pdf) .

*** DATA: Communications Earth & Environment endorses the principles of the Enabling FAIR data project (<http://www.copdess.org/enabling-fair-data-project/>). We ask authors to make the data that support their conclusions available in permanent, publically accessible data repositories. (Please contact the editor if you are unable to make your data available).

All Communications Earth & Environment manuscripts must include a section titled "Data Availability" at the end of the Methods section or main text (if no Methods). More information on this policy, is available at Data Availability

<http://www.nature.com/authors/policies/data/data-availability-statements-data-citations.pdf>><http://www.nature.com/authors/policies/data/data-availability-statements-data-citations.pdf>.

DATA SOURCES: All new data associated with the paper should be placed in a persistent repository where they can be freely and enduringly accessed. We recommend submitting the data to discipline-specific, community-recognized repositories, where possible and a list of recommended repositories is provided at ><http://www.nature.com/sdata/policies/repositories>.

If a community resource is unavailable, data can be submitted to generalist repositories such as https://figshare.com/>figshare or http://datadryad.org/>Dryad Digital Repository. Please provide a unique identifier for the data (for example a DOI or a permanent URL) in the data availability statement, if possible. If the repository does not provide identifiers, we encourage authors to supply the search terms that will return the data. For data that have been obtained from publically available sources, please provide a URL and the specific data product name in the data availability statement. Data with a DOI should be further cited in the methods reference section.

Please refer to our data policies at ><http://www.nature.com/authors/policies/availability.html>.

REVIEWER COMMENTS:

Reviewer #1 (Remarks to the Author):

The manuscript is interesting to read and will attract a wider scientific audience since it is, indeed, the first attempt to date 350,000-year recharge history in the southern Great Basin. However, I disagree with authors about their statement (line 13: “This study provides the first precisely dated...”). I don’t think this is an appropriate way to start this paper, because due to uncertainties (which were expected due to the nature of the work and uncertainties also indicated by authors in the manuscript) and overall approach, we can’t be sure it is “precisely dated” at all. Maybe it is, but we can’t know for sure on the basis of information provided. It is certainly a very good attempt to get a solid estimation of the recharge history, so I would remove “precisely” from the text in that line, as well as any other similar attempts in text to oversell the manuscript (authors have already presented a very good and scientifically sound

work, and overselling is in my opinion not necessary and harmful), and focus on what has been supported by their results.

I definitely agree with authors about the novelty of the approach, and I enjoyed reading about historical changes in water table in the study area. That being said, this paper will likely be a significant contribution and can be recommended for publication, although, some minor changes should be made. For example, model limitations were very well explained but those facts should be incorporated in other parts of the text (mentioning limitations in abstract, at least, very briefly is necessary, as abstract provides no insights into limitations of the proposed approach and those effects on results).

The scientific quality is very good and paper was easy to read and follow. No parts of manuscript were too long or too detailed, and that makes reading this paper enjoyable. Figures are produced to a high standard, however, I must suggest some changes in Figure 2. Firstly, I would split the legend into two rows so the figure looks nicer and the legend won't block any part of the graph, and secondly, I wouldn't use green and red colour on the same graph (not only because of colour blind readers but also because they are too close to each other on visible spectrum which is unpleasant to our eyes; thus, changing these red triangles into any other colour shouldn't be too difficult for authors).

Reviewer #2 (Remarks to the Author):

Review of Jackson et al "A 350,000-year history of groundwater recharge in the southern Great Basin, USA"

Summary

Jackson and colleagues model groundwater recharge in the southern Great Basin using the recently developed water table reconstruction from Devil's Hole. This work is important and the paper is well written. My main comments are below, but in particular I found the comparison to geologic data in any sort of quantitative (spatial) way very much lacking (literally circles on a map) and the final analysis of the paper, switching to a transient simulation and commenting on the future of the DH pupfish vastly underdeveloped (unlike the steady-state modeling), and thus I suggest removing from the paper.

I suggest major revisions and upon revision would be supportive of this paper's eventual publication. The text and figures are thorough and well written, and I appreciate the goals of this paper to develop this modeling framework.

Major Comments

Abstract - needs to be much more specific, as written details are lost. Suggestions below.

117-131: But glacial-interglacial cycles show abrupt change on <10 kyrs timescales (e.g., the last deglaciation), I'm not sure that I understand the reasoning behind the study design here. A fully transient model for ~350 ka is what the title and description implied and would be needed.

156-171: So the recharge changes are scaled uniformly across the domain? Given the topographic complexity this seems unlikely to me especially as the atmosphere cools the lapse rates will change. Perhaps using the PMIP3/PMIP4 LGM and LIG model simulations would have been helpful here to validate this assumption. Oster et al. (2015) was focused on precip changes, the P-ET maps in their supplement look quite different. Further, newer climate models are now available. When this is brought up again in 312-313, it would maybe help to annotate this on one of the axes of Figure 3.

237-238: How does this compare to actual geologic data of the distribution of tufa/spring/travertine deposits spatial? I was surprised not to see this as further validation of the modeling efforts. The "area of mapped paleogroundwatre discharge" shown in circles is really not adequate for comparison, is it possible to digitize the previous studies mentioned in 273-277?

288-290: In general I found that this broad sweeping statement was not adequately supported by the data-model comparison in this study.

292-296: The authors should look at opal based (234U-238U) initial infiltration rates from Maher et al. (2014, AJS), they also used the Yucca Mountain data.

324-332: The sudden switch to a transient model here seems out of scope to me, I would suggest moving this out of the paper and writing a separate (management focused paper) as this needs much more fleshing out and uncertainty quantification.

Same for 361-369, this is overstepping what a majority of the paper was about and what was modeled.

Much more is needed to make such strong statements w.r.t. future groundwater and the DH pufish.

Specific Line-by-Line Comments

13-15: the phrasing here sounds like the water-table record is new in this work, which it is not.

16: which interglacial?

17: which glacial

20: error bars on these slopes?

22: either "approximately four times" or just state that it's 4.2 times more sensitivity.

23-24: using which/what CMIP scenarios or otherwise?

45-50: How about precip trends? Also I would suggest you cite trends in the newer AR6/CMIP6 scenarios, RCP8.5 is now ~10 years old.

Figure 1. Cite groundwater basin boundary reference in caption, also how was this determined/modified?

59: This is true but not in the Ash Meadows region based on maps by Reheis and others.

67-69: I would suggest you also cite Matsubara and Howard (2009) and Ibarra et al. (2018) here, who actually looked at lake areas and the regional water balance.

76-79: True but from my knowledge of the literature there is literally no other archives as unique as DH.

96: "at" before "steady"

151-155: Where did all the modern heads come from? are they all pre-1950 as implied above?

172-179: it would have been helpful to show these lakes on Figure 1

Figure 3: Change color of the modern lines

263: Do we know the slip rate?

322: driven by temperature and precip trends or just temperature (via evaporation)?

339: Again make clear there is no new DH water level data reported here.

Reviewer #3 (Remarks to the Author):

Jackson et al take the water level reconstructions at Devils Hole by Wendt et al. (2018) and Szabo et al (1994) a step further by modeling how recharge has changed over time. This is an important study that highlights the sensitivity to water table fluctuations in the southern Great Basin in response to changes in paleoclimate. This retrospective view of recharge, anchored by the water level reconstructions, thus has implications of future change, given that the slope of water table and recharge relations can be constrained for past interglacial conditions. The model was tested, in part, by simulating areas of discharge under previous pluvial periods and matching those to geologic data.

The main strength of this paper is assigning quantitative recharge estimates to the water level data in Devils Hole. While I cannot comment on the groundwater flow model, the authors seem to have done a nice job constraining the modern and past system. One weakness of the current version of the manuscript, though, is assessing how low water tables might go in the future in the face of climate change, because the available data do not seem to capture previous periods of enhanced aridity relative to today.

To elaborate, the authors mention sensitivity of water levels to millennial-scale fluctuations. The middle Holocene is thus an important target, given the widespread evidence of winter dryness in the region. This time period is also a potential analog for the future, given that aridity more extreme than today seems to have lasted for at least a few millennia. My first question is, do the dated speleothems constrain the middle Holocene dryness in the region? The Wendt et al data suggests the absence of folia in the -1.6 m core (n =1) as evidence that water levels never dropped below this level. However, there are at least two potential possibilities and/or implications of this lack of folia at -1.6 m. The first is the possibility that the single core missed folia at -1.6 m when they might be found elsewhere at the same or even lower levels (given that an n=1 is not a large sample size). Folia in some caves are not always continuous in space (D'Angeli et al., 2015). Another possibility is that water tables lower than -1.6 m were present but too brief to form folia.

The major implication is, assuming that there were no folia, at any time and anyplace in Devils at -1.6 m and below, as suggested by Wendt et al, why would the water table be insensitive to the Middle Holocene aridity on millennial timescales? I.e., could there be a water table 'floor' to climate more arid than today, perhaps set by the altitude of subsurface impermeable barriers or faults? The reason this is important is because the paper pushes into forecasting the future based on climate change, yet the dated speleothems only provide a retrospective view. If there is indeed a 'floor' below which the water table will not fall, then the future projections (and concern about the pupfish) may be overestimated. Alternatively, if a few millennia of aridity are not enough to lower the water table in the past, then the

water table may actually be more stable in the next few millennia, too. A few relevant references for Middle Holocene drying include (Grayson, 2000; Grayson, 2011; Lachniet et al., 2020; Louderback et al., 2010).

Like any modeling study, there are additional assumptions, which all seem reasonable enough, at least for a first pass through of assessing recharge variations related to water level fluctuations.

Lines 101-102, if 80% of water is sourced in spring and sheep mountains as based on the modern, this proportion could have changed during past cold (or even warm, i.e., altithermal) conditions. If the proportion has changes significantly, then future projections need to account for that too. The response of the water table in the future might also relate to where (geographically speaking) the precipitation falls within the GW basin. How do the authors assess this?

How does the gw flow model's results change if the more northern interbasin flow areas are included, i.e., Railroad Valley (Rose and Davisson, 2003)? How might their importance change relative to the AM basin for pluvial vs. interpluvial periods? Add something here to lines 248-250, which acknowledges this uncertainty a little bit, but doesn't engage with it in enough detail.

Some additional data and perspective is needed in the water level reconstruction. How precisely are the speleothems recording the water table? What is the accuracy of the surveying within the cave? Include those uncertainties in the graphs. A few additional publications on folia that give wider ranges than the assumption that folia are just formed at the water table: (D'Angeli et al., 2015; Hill and Forti, 1997; Szabo et al., 1994)

It would also be worth mentioning the issues of ages and age uncertainty on the Devils Hole speleothems, given the published data showing excess ^{230}Th in the water column. Figure 2 has age uncertainties plotted, but it isn't clear if these are just analytical or also take into account uncertainty in initial thorium, and this should be discussed in the paper.

Finally, I'm curious about the explanation for the curvilinear water table/recharge relationship. Why?

Citations

D'Angeli, I. M., De Waele, J., Melendres, O. C., Tisato, N., Sauro, F., Gonzales, E. R. G., Bernasconi, S. M., Torriani, S., and Bontognali, T. R. R., 2015, Genesis of folia in a non-thermal epigenic cave (Matanzas, Cuba): *Geomorphology*, v. 228, p. 526-535.

Grayson, D. K., 2000, Mammalian Responses to Middle Holocene Climatic Change in the Great Basin of the Western United States: *Journal of Biogeography*, v. 27, no. 1, p. 181-192.

Grayson, D. K., 2011, *The Great Basin: a Natural Prehistory*, University of California Press, 418 p.:

Hill, C., and Forti, P., 1997, *Cave minerals of the world*, Huntsville, Alabama, National

Speleological Society, 463 p.:

Lachniet, M. S., Asmerom, Y., Polyak, V., and Denniston, R., 2020, Great Basin Paleoclimate and Aridity Linked to Arctic Warming and Tropical Pacific Sea Surface Temperatures: *Paleoceanography and Paleoclimatology*, v. 35, no. 7, p. e2019PA003785.

Louderback, L. A., Grayson, D. K., and Llobera, M., 2010, Middle-Holocene climates and human population densities in the Great Basin, western USA: *The Holocene*, v. 21, no. 2, p. 366-373.

Rose, T. P., and Davisson, M. L., 2003, Isotopic and geochemical evidence for Holocene-age groundwater in regional flow systems of south-central Nevada: *Special Paper - Geological Society of America*, v. 368, p. 143-164.

Szabo, B. J., Kolesar, P. T., Riggs, A. C., Winograd, I. J., and Ludwig, K. R., 1994, Paleoclimatic inferences from a 120,000-yr calcite record of water-table fluctuation in Browns Room of Devils Hole, Nevada: *Quaternary Research (New York)*, v. 41, no. 1, p. 59-69.

Wendt, K. A., Pythoud, M., Moseley, G. E., Dublyansky, Y. V., Edwards, R. L., and Spötl, C., 2019, Paleohydrology of southwest Nevada (USA) based on groundwater $^{234}\text{U}/^{238}\text{U}$ over the past 475 k.y: *GSA Bulletin*.

REVIEWER COMMENTS:

Reviewer #1 (Remarks to the Author):

The manuscript is interesting to read and will attract a wider scientific audience since it is, indeed, the first attempt to date 350,000-year recharge history in the southern Great Basin. However, I disagree with authors about their statement (line 13: "This study provides the first precisely dated..."). I don't think this is an appropriate way to start this paper, because due to uncertainties (which were expected due to the nature of the work and uncertainties also indicated by authors in the manuscript) and overall approach, we can't be sure it is "precisely dated" at all. Maybe it is, but we can't know for sure on the basis of information provided. It is certainly a very good attempt to get a solid estimation of the recharge history, so I would remove "precisely" from the text in that line, as well as any other similar attempts in text to oversell the manuscript (authors have already presented a very good and scientifically sound work, and overselling is in my opinion not necessary and harmful), and focus on what has been supported by their results.

Answer: Removed "precisely" from text throughout the manuscript.

I definitely agree with authors about the novelty of the approach, and I enjoyed reading about historical changes in water table in the study area. That being said, this paper will likely be a significant contribution and can be recommended for publication, although, some minor changes should be made. For example, model limitations were very well explained but those facts should be incorporated in other parts of the text (mentioning limitations in abstract, at least, very briefly is necessary, as abstract provides no insights into limitations of the proposed approach and those effects on results).

Answer: Briefly mentioned model uncertainties in abstract and summarized model limitations (uncertainties) in summary and conclusions section.

The scientific quality is very good and paper was easy to read and follow. No parts of manuscript were too long or too detailed, and that makes reading this paper enjoyable. Figures are produced to a high standard, however, I must suggest some changes in Figure 2. Firstly, I would split the legend into two rows so the figure looks nicer and the legend won't block any part of the graph, and secondly, I wouldn't use green and red colour on the same graph (not only because of colour blind readers but also because they are too close to each other on visible spectrum which is unpleasant to our eyes; thus, changing these red triangles into any other colour shouldn't be too difficult for authors).

Answer: Splitting the legend into two columns is a great suggestion. We changed the green tone - for me (TRJ) as a colour-blind person and on my screen the colour scheme works great. We'd like to keep blue (cold) and red (warm).

Reviewer #2 (Remarks to the Author):

Review of Jackson et al "A 350,000-year history of groundwater recharge in the southern Great Basin, USA"

Summary

Jackson and colleagues model groundwater recharge in the southern Great Basin using the recently developed water table reconstruction from Devil's Hole. This work is important and the paper is well written. My main comments are below, but in particular I found the comparison to geologic data in any sort of quantitative (spatial) way very much lacking (literally circles on a map) and the final analysis of the paper, switching to a transient simulation and commenting on the future of the DH pupfish vastly underdeveloped (unlike the steady-state modeling), and thus I suggest removing from the paper. I suggest major revisions and upon revision would be supportive of this paper's eventual publication. The text and figures are thorough and well written, and I appreciate the goals of this paper to develop this modeling framework.

Answer: See responses to these major comments below.

Major Comments

Abstract - needs to be much more specific, as written details are lost. Suggestions below.

Answer: See responses to suggested comments below.

117-131: But glacial-interglacial cycles show abrupt change on <10 kyrs timescales (e.g., the last deglaciation), I'm not sure that I understand the reasoning behind the study design here. A fully transient model for ~350 ka is what the title and description implied and would be needed.

Answer: Thank you for this comment. We were not clear about what was fully meant by the quote: "The time to equilibrate to a steady-state condition is within 10,000 years". Transient model simulations were run to determine approximate timescales for equilibration to steady-state conditions. The 10,000 years is a maximum equilibration limit that is unlikely in the real system. During equilibration from one recharge state to another (e.g., modern to 20% decrease in recharge), water-level changes follow an exponential trend. A large part (80–90%) of the water-level change during equilibration occurs within the first 1,000 years, and the long exponential equilibration timescale (between 5,000 and 10,000 years) occurs for full equilibration to small water-level changes of less than 0.1 m. Furthermore, these transient models simulate abrupt changes in recharge, whereas natural climate fluctuations will result in gradual recharge changes with time, such that maximum magnitudes of water-level change between sequential climate states will be small within 1,000 years.

156-171: So the recharge changes are scaled uniformly across the domain? Given the topographic complexity this seems unlikely to me especially as the atmosphere cools the lapse rates will change. Perhaps using the PMIP3/PMIP4 LGM and LIG model simulations would have been helpful here to validate this assumption.

Answer: Assuming the spatial distribution of recharge remained constant during the last 350,000 years is reasonable. Land-surface topography was relatively stable during this time because the mountains were already formed. Saturated hydraulic properties also are relatively stable. If the topography is the same, then the recharge distribution is still controlled by high and low topographic

altitudes, despite atmospheres that are either cooler and wetter or warmer and drier. Therefore, the basic pattern of high and low recharge will not change because the recharge pattern accounts for the topographic complexity of the study area, which has been relatively stable. Cooler and wetter versus warmer and drier climates are simulated with the application of a recharge multiplier. A recharge multiplier is a reasonable extrapolation based on available information and the reasonable assumption of a stable topographic setting. Based on a literature search, we have not seen other climate models that account for geologic structure changes, such as mountain-block formation and erosion, in simulating past climates. Upon further thought, this is a valid assumption, not a model limitation, and has been removed from the model limitations section.

Oster et al. (2015) was focused on precip changes, the P-ET maps in their supplement look quite different. Further, newer climate models are now available. When this is brought up again in 312-313, it would maybe help to annotate this on one of the axes of Figure 3.

Answer: Even though the P-ET maps in supplemental figure S2 of Oster et al. (2015) look quite different in general, the range of P-ET is between 200 and 300% for the study area (southern Nevada). This range refers to the models that show the strongest agreement with the proxy network (CNRM-CM5, IPSL-CM5A-LR, MPI-ESM-P, CCSM4 and MIROC-ESM). The MIROC-ESM model is an outlier. There have been updates to the climate models used by Oster et al. (2015) in the past 7 years. However, we are not aware of a study comparable to Oster et al. (2015) that uses these updated models. Since Figure 3 has no time dimension, adding the numbers discussed in these lines would be an oversimplification.

237-238: How does this compare to actual geologic data of the distribution of tufa/spring/travertine deposits spatial? I was surprised not to see this as further validation of the modeling efforts. The "area of mapped paleo-groundwater discharge" shown in circles is really not adequate for comparison, is it possible to digitize the previous studies mentioned in 273-277?

Answer: Figure 4 has been modified to show digitized areas of paleo-groundwater discharge from previous work.

288-290: In general I found that this broad sweeping statement was not adequately supported by the data-model comparison in this study.

Answer: Authors respectfully disagree. Areas of simulated groundwater discharge generally follow mapped areas of paleo-discharge deposits. The model cannot be calibrated to the exact paleo-discharge locations because paleo-water-table data only are available at one location: Devils Hole.

292-296: The authors should look at opal based (234U-238U) initial infiltration rates from Maher et al. (2014, AJS), they also used the Yucca Mountain data.

Answer: Thank you for suggestion. Added Maher et al. (2014) results to manuscript.

324-332: The sudden switch to a transient model here seems out of scope to me, I would suggest moving this out of the paper and writing a separate (management focused paper) as this needs

much more fleshing out and uncertainty quantification. Same for 361-369, this is overstepping what a majority of the paper was about and what was modeled. Much more is needed to make such strong statements w.r.t. future groundwater and the DH pupfish.

Answer: Deleted section on transient model and results in summary and conclusions section.

Specific Line-by-Line Comments

13-15: the phrasing here sounds like the water-table record is new in this work, which it is not.

Answer: Text revised for clarity.

16: which interglacial? 17: which glacial

Answer: This refers to the peak (i.e., most extreme) conditions during the last 350,000 years (as apparent from the title and previous sentence). We see the reviewer's point, but in order to keep it simple and to focus on the important points we restrain from adding a more specific time component here. Peak interglacial conditions occurred during MIS 5, and peak glacial conditions are documented 3 times (summarized in Figure 2).

20: error bars on these slopes?

Answer: No. Unquantifiable uncertainties associated with model effort. Only qualitative comparisons can adequately validate the modelling effort.

22: either "approximately four times" or just state that it's 4.2 times more sensitivity.

Answer: Added "about"

23-24: using which/what CMIP scenarios or otherwise?

Answer: Deleted.

45-50: How about precip trends? Also I would suggest you cite trends in the newer AR6/CMIP6 scenarios, RCP8.5 is now ~10 years old.

Answer: Updated text regarding temperature and precipitation trends using AR6/CMIP6 projections. Updated references to IPCC (2021; 2022).

Figure 1. Cite groundwater basin boundary reference in caption, also how was this determined/modified?

Answer: Citation is in explanation, so no need to cite again in the caption. The DV3 model uses the boundaries delineated in figure 1. The original source for the boundaries is Fenelon et al. (2016). The Fenelon et al. work was being done simultaneously with, but independently of, the DV3 model effort. An older version of the boundaries was used for the DV3 model domain and minor revisions were made to the basin boundaries after the DV3 model was developed. If a reader looked closely at Fenelon et al. (2016; plate 1) and DV3 model domain boundary, nuanced differences can be discerned. We cited the original source for the boundaries. This information is not relevant for publishing.

59: This is true but not in the Ash Meadows region based on maps by Reheis and others.

Answer: This is true for the Alkali Flat–Furnace Creek groundwater basin that is part of the study area.

67-69: I would suggest you also cite Matsubara and Howard (2009) and Ibarra et al. (2018) here, who actually looked at lake areas and the regional water balance.

Answer: Added these references. Thank you.

76-79: True but from my knowledge of the literature there is literally no other archives as unique as DH.

Answer: Devils Hole is a unique setting that provides information on paleo-water-table fluctuations. However, other types of paleo data, such as mapped paleo-discharge areas, can be used with a similar approach. For example, a groundwater model can be developed where hydraulic properties are estimated using modern water level and discharge data. Then, when the modern groundwater model is converted into a superposition model, simulated water-level change maps can be produced to provide an understanding of the spatial distribution of water-level changes that would be needed to form paleo-discharge areas that formed under wetter climatic conditions.

96: "at" before "steady"

Answer: Corrected.

151-155: Where did all the modern heads come from? are they all pre-1950 as implied above?

Answer: The spatial distribution of modern heads was obtained from the calibrated DV3-SS model (Halford and Jackson, 2020). These modern heads are pre-1950 heads. Text revised for clarity.

172-179: it would have been helpful to show these lakes on Figure 1

Answer: Added pluvial lake areas to figure 1.

Figure 3: Change color of the modern lines

Answer: Color scheme of Fig 3 corresponds to Fig 2. For Fig. 2 it was adjusted based to other comments. Red is for the discussed warm (interglacial) climate state.

263: Do we know the slip rate?

Answer: Maximum vertical offset of 0.6 m. Added to manuscript.

322: driven by temperature and precip trends or just temperature (via evaporation)?

Answer: Deleted section.

339: Again make clear there is no new DH water level data reported here.

Answer: Revised for clarity.

Reviewer #3 (Remarks to the Author):

Jackson et al take the water level reconstructions at Devils Hole by Wendt et al. (2018) and Szabo et al (1994) a step further by modeling how recharge has changed over time. This is an important study that highlights the sensitivity to water table fluctuations in the southern Great Basin in response to changes in paleoclimate. This retrospective view of recharge, anchored by the water level reconstructions, thus has implications of future change, given that the slope of water table and recharge relations can be constrained for past interglacial conditions. The model was tested, in part, by simulating areas of discharge under previous pluvial periods and matching those to geologic data.

The main strength of this paper is assigning quantitative recharge estimates to the water level data in Devils Hole. While I cannot comment on the groundwater flow model, the authors seem to have done a nice job constraining the modern and past system. One weakness of the current version of the manuscript, though, is assessing how low water tables might go in the future in the face of climate change, because the available data do not seem to capture previous periods of enhanced aridity relative to today.

To elaborate, the authors mention sensitivity of water levels to millennial-scale fluctuations. The middle Holocene is thus an important target, given the widespread evidence of winter dryness in the region. This time period is also a potential analog for the future, given that aridity more extreme than today seems to have lasted for at least a few millennia. My first question is, do the dated speleothems constrain the middle Holocene dryness in the region? The Wendt et al data suggests the absence of folia in the -1.6 m core (n =1) as evidence that water levels never dropped below this level. However, there are at least two potential possibilities and/or implications of this lack of folia at -1.6 m. The first is the possibility that the single core missed folia at -1.6 m when they might be found elsewhere at the same or even lower levels (given that an n=1 is not a large sample size). Folia in some caves are not always continuous in space (D'Angeli et al., 2015). Another

possibility is that water tables lower than -1.6 m were present but too brief to form folia.

Answer: The DH water table appears to be in continuous decline from glacial highs during the mid-Holocene dry period (circa 6 ka). However, due to low data resolution, we are unable to resolve short-term fluctuations during this time. There is no obvious folia deposition in Devils Hole 2 below +0m during this time interval, but—as the reviewer pointed out—it's possible that a brief decrease below +0m may have occurred. Data from Devils Hole #2 cave below +0m are currently unavailable for the entirety of the Holocene. Because of this lack of continuity, we are unable to confidently constrain the water-table elevation during the mid-Holocene.

Instead, this study examines the last interglacial (MIS5e), during which the Great Basin was potentially warmer and drier than the mid-Holocene due to insolation forcings. The core at -1.6 m documents continuous mammillary calcite deposition during the entirety of MIS 5e. We are therefore confident that the water table never fell below -1.6 m during this period. Due to firmer data constraints, we have chosen to focus on the MIS 5e dry period.

The major implication is, assuming that there were no folia, at any time and anyplace in Devils at -1.6 m and below, as suggested by Wendt et al, why would the water table be insensitive to the Middle Holocene aridity on millennial timescales? I.e., could there be a water table 'floor' to climate more arid than today, perhaps set by the altitude of subsurface impermeable barriers or faults? The reason this is important is because the paper pushes into forecasting the future based

on climate change, yet the dated speleothems only provide a retrospective view. If there is indeed a 'floor' below which the water table will not fall, then the future projections (and concern about the pupfish) may be overestimated. Alternatively, if a few millennia of aridity are not enough to lower the water table in the past, then the water table may actually be more stable in the next few millennia, too. A few relevant references for Middle Holocene drying include (Grayson, 2000; Grayson, 2011; Lachniet et al., 2020; Louderback et al., 2010).

Answer: The section about future projections was removed from the paper. The Middle Holocene is addressed in the previous comment. There is a water-table "floor". The rate of water-table change below modern conditions, as shown on Figure 3, is small because nearby Ash Meadows springs are upholding the water table. Once the springs go dry, the water table will drastically drop to a minimum of 800 m below modern conditions, which is the water-table difference between Devils Hole and the Death Valley floor. The Death Valley floor is the terminus of the groundwater system. When all groundwater-discharge areas go dry, the water table will drop and hydraulic gradients will approach zero as water levels equilibrate to a global minimum defined by the groundwater terminus.

Like any modeling study, there are additional assumptions, which all seem reasonable enough, at least for a first pass through of assessing recharge variations related to water level fluctuations.

Lines 101-102, if 80% of water is sourced in spring and sheep mountains as based on the modern, this proportion could have changed during past cold (or even warm, i.e., altithermal) conditions. If the proportion has changes significantly, then future projections need to account for that too. The response of the water table in the future might also relate to where (geographically speaking) the precipitation falls within the GW basin. How do the authors assess this?

Answer: We realized that future projections are beyond the scope of the paper, based on reviewer #2 comments, and have removed them from the paper.

How does the gw flow model's results change if the more northern interbasin flow areas are included, i.e., Railroad Valley (Rose and Davisson, 2003)? How might their importance change relative to the AM basin for pluvial vs. interpluvial periods? Add something here to lines 248-250, which acknowledges this uncertainty a little bit, but doesn't engage with it in enough detail.

Answer: Expanding the model domain to include areas such as Railroad Valley would have no effect on the water levels in Devils Hole. Interbasin flow from Railroad Valley and other bounding basins is negligible under modern conditions. During wetter periods, groundwater-flow paths shorten between recharge and discharge areas, and more discharge areas form; therefore, the contributing area to Devils Hole will be smaller. During drier periods, compared to modern, modern discharge areas will either decrease in areal extent, remain the same size but have decreased discharge rates, or disappear due to lowered water levels. Given the purpose of the model effort is to estimate recharge volumes, based on the contributing area to Devils Hole, interbasin flow is a non-issue.

Some additional data and perspective is needed in the water level reconstruction. How precisely are the speleothems recording the water table? What is the accuracy of the surveying within the cave? Include those uncertainties in the graphs. A few additional publications on folia that give wider ranges than the assumption that folia are just formed at the water table: (D'Angeli et al., 2015; Hill and Forti, 1997; Szabo et al., 1994)

Answer: We used the water-table reconstruction, as published by Wendt et al. (2018), which agrees with the earlier work by Szabo et al. (1994). The other aspects and details are part of the methodology of Wendt et al. and described there. We added a statement to the paper referring to the more detailed discussion (in Wendt et al., 2018) of the valid points raised with this comment.

It would also be worth mentioning the issues of ages and age uncertainty on the Devils Hole speleothems, given the published data showing excess ^{230}Th in the water column. Figure 2 has age uncertainties plotted, but it isn't clear if these are just analytical or also take into account uncertainty in initial thorium, and this should be discussed in the paper.

Answer: We understand the point and addressed it by referring to previous work (Wendt et al., 2018) in the manuscript where this question is discussed in detail. Our study does not change anything of the chronology of Wendt et al (2018).

Finally, I'm curious about the explanation for the curvilinear water table/recharge relationship. Why?

Answer: The rate of water-table change below modern conditions is small because nearby Ash Meadows springs are upholding the water table. Once the springs go dry, the water table will drastically drop to a minimum of 800 m below modern conditions, which is the water-table difference between Devils Hole and the Death Valley floor. Conversely, the rate of water-table change above modern conditions steepens with increasing recharge because more water is needed to elevate the water table over a large spatial area since wetter conditions form more discharge areas that stabilize the water table.

Citations

D'Angeli, I. M., De Waele, J., Melendres, O. C., Tisato, N., Sauro, F., Gonzales, E. R. G., Bernasconi, S. M., Torriani, S., and Bontognali, T. R. R., 2015, Genesis of folia in a non-thermal epigenic cave (Matanzas, Cuba): *Geomorphology*, v. 228, p. 526-535.

Grayson, D. K., 2000, Mammalian Responses to Middle Holocene Climatic Change in the Great Basin of the Western United States: *Journal of Biogeography*, v. 27, no. 1, p. 181-192.

Grayson, D. K., 2011, *The Great Basin: a Natural Prehistory*, University of California Press, 418 p.:

Hill, C., and Forti, P., 1997, *Cave minerals of the world*, Huntsville, Alabama, National Speleological Society, 463 p.:

Lachniet, M. S., Asmerom, Y., Polyak, V., and Denniston, R., 2020, Great Basin Paleoclimate and Aridity Linked to Arctic Warming and Tropical Pacific Sea Surface Temperatures: *Paleoceanography and Paleoclimatology*, v. 35, no. 7, p. e2019PA003785.

Louderback, L. A., Grayson, D. K., and Llobera, M., 2010, Middle-Holocene climates and human population densities in the Great Basin, western USA: *The Holocene*, v. 21, no. 2, p. 366-373.

Rose, T. P., and Davisson, M. L., 2003, Isotopic and geochemical evidence for Holocene-age

groundwater in regional flow systems of south-central Nevada: Special Paper - Geological Society of America, v. 368, p. 143-164.

Szabo, B. J., Kolesar, P. T., Riggs, A. C., Winograd, I. J., and Ludwig, K. R., 1994, Paleoclimatic inferences from a 120,000-yr calcite record of water-table fluctuation in Browns Room of Devils Hole, Nevada: Quaternary Research (New York), v. 41, no. 1, p. 59-69.

Wendt, K. A., Pythoud, M., Moseley, G. E., Dublyansky, Y. V., Edwards, R. L., and Spötl, C., 2019, Paleohydrology of southwest Nevada (USA) based on groundwater $^{234}\text{U}/^{238}\text{U}$ over the past 475 k.y: GSA Bulletin.

1st Nov 22

Dear Mr Steidle,

Please allow me to sincerely apologise for the long delay you have experienced waiting for a decision on your manuscript titled "A 350,000-year history of groundwater recharge in the southern Great Basin, USA". It has now been seen again by 3 reviewers, and I include their comments at the end of this message. Reviewers #2 and #3 are the same as in the last round but Reviewer #1 did not submit another report and Reviewer #4 is new this round.

They find your work of interest, but some important points are raised, particularly by Reviewer #4 with respect to the hydrological modelling itself. We remain interested in the possibility of publishing your study in *Communications Earth & Environment*, but would like to consider your responses to these concerns and assess a revised manuscript before we make a final decision on publication.

We therefore invite you to revise and resubmit your manuscript, along with a point-by-point response that takes into account the points raised and indicates where the relevant changes have been made in the text itself. Please highlight all changes in the manuscript text file. Specifically, in response to the concerns of Reviewer #4 we request that you fully explain and justify your methodology, including alterations and assumptions used in the modeling approach, to ensure that the hydrological modelling is robust.

Please use the following link to submit your revised manuscript, point-by-point response to the referees' comments (which should be in a separate document to any cover letter) and the completed checklist:

[link redacted]

We hope to receive your revised paper within six weeks; please let us know if you aren't able to submit it within this time so that we can discuss how best to proceed. If we don't hear from you, and the revision process takes significantly longer, we may close your file. In this event, we will still be happy to reconsider your paper at a later date, as long as nothing similar has been accepted for publication at *Communications Earth & Environment* or published elsewhere in the meantime.

We understand that due to the current global situation, the time required for revision may be longer than usual. We would appreciate it if you could keep us informed about an estimated timescale for resubmission, to facilitate our planning. Of course, if you are unable to estimate, we are happy to accommodate necessary extensions nevertheless.

Please do not hesitate to contact me if you have any questions or would like to discuss these revisions further. We look forward to seeing the revised manuscript and thank you for the

opportunity to review your work.

Best regards,

Joe Aslin

Senior Editor,
Communications Earth & Environment
<https://www.nature.com/commsenv/>
Twitter: @CommsEarth

EDITORIAL POLICIES AND FORMATTING

Editorial Policy: [Policy requirements](https://www.nature.com/documents/nr-editorial-policy-checklist.pdf) (Download the link to your computer as a PDF.)

Furthermore, please align your manuscript with our format requirements, which are summarized on the following checklist:

[Communications Earth & Environment formatting checklist](https://www.nature.com/documents/commsj-phys-style-formatting-checklist-article.pdf)

and also in our style and formatting guide [Communications Earth & Environment formatting guide](https://www.nature.com/documents/commsj-phys-style-formatting-guide-accept.pdf) .

***** DATA:** Communications Earth & Environment endorses the principles of the Enabling FAIR data project (<http://www.copdess.org/enabling-fair-data-project/>). We ask authors to make the data that support their conclusions available in permanent, publically accessible data repositories. (Please contact the editor if you are unable to make your data available).

All Communications Earth & Environment manuscripts must include a section titled "Data Availability" at the end of the Methods section or main text (if no Methods). More information on this policy, is available at <http://www.nature.com/authors/policies/data/data-availability-statements-data-citations.pdf>.

If a community resource is unavailable, data can be submitted to generalist repositories such as [figshare](https://figshare.com/) or [Dryad Digital Repository](http://datadryad.org/). Please provide a unique identifier for the data (for example a DOI or a permanent URL) in the data availability statement, if possible. If the repository does not provide identifiers, we encourage authors to supply the search terms that will return the data. For data that have been obtained from publically available sources, please provide a URL and the specific data product name in the data availability statement. Data with a DOI should be further cited in the methods reference section.

REVIEWER COMMENTS:

Reviewer #2 (Remarks to the Author):

I re-read the paper and rebuttal document from Jackson et al. and largely agree with their changes. I appreciate the thoughtful responses. Three minor points:

- 1) I still don't find figure 4's comparison to geologic data particularly satisfying, if the model was representing reality then a more quantitative comparison to the spatial data should be possible. That said, I understand that this is not the point of the paper.
- 2) Further the inability to place some level of uncertainty on the estimates based on model assumptions is somewhat unsatisfying, but again I understand how complex it is to run such models, and thus such a sensitivity test can be left for another paper.
- 3) Finally, the rebuttal to why a recharge multiple is convincing, but what about a uniform increase in recharge (in units of say mm/yr) across the domain? Would that be more realistic? If you think about Clausius–Clapeyron scaling of the water cycle (likely that is too sensitive) delivery of water to a region under warmer or colder periods should scale by amount not a multiple right?

I make a few final (and very minor suggestions below) related to changes and look forward to future group from this group on the (transient) modeling of this system.

Minor comments/changes:

121-123: You may wish to note though that some of the work since Wendt 2018 (Wendt et al., 2020 and Li et al., 2020) show systematic changes in systems such as the $(^{234}\text{U}/^{238}\text{U})_{\text{initial}}$ (also denoted $d_{234}\text{U}$) that provide a more 'continuous' history of recharge, albeit with some delay and geochemical complications

193: change to climate models or global climate models

Map figures: these are somewhat grainy and the text pretty small for this size of a figure. Consider adjusting for publication.

Reviewer #3 (Remarks to the Author):

Overall this remains an important paper. I do feel there were some lost opportunities in the revision to handle some of the reviewers' comments.

Some points seemed to mainly be argued in the response to reviewers rather than modifying the manuscript to account for the suggestions. One example is the reviewer question beginning "117-131: But glacial-interglacial cycles show abrupt changes on <10 kyrs timescales..." How did the authors modify the manuscript to account for this important observation?

Another is the evaluation of data relating to the question of Mid Holocene aridity. Was the manuscript modified to account for this important point? If the DH folio do not record the enhance mid-Holocene aridity then one has to question how well they were able to account for other millennial-scale aridity intervals. What I would like to see is how the manuscript was modified to account for this point, but I did not see where in the manuscript this was handled. Nor how the manuscript was modified to address the issue of interbasin flow from Railroad Valley and other locations. At the least it is necessary to state in the manuscript that a lack of interbasin flow into the modeled basins from further north is an assumption.

Reviewer #4 (Remarks to the Author):

The authors take a novel approach to simulating potential recharge values that could explain estimated paleo water table levels throughout a 350k yr span. The authors converted an existing groundwater model for the AMGB to a superposition model to analyze changes from current, baseline values. The model design was such that relative changes in water level under different recharge values ranging from -24-300% from current estimated recharge across the basin were analyzed. The method was to develop a relationship between recharge and expected water level and determine what recharge would generate observed water levels at a single point in Devils Hole where a paleo-water table record existed.

The model is unique, and I think the study and results are intriguing. The article goes out of its way to avoid too much modelling jargon. However, as I address later, it also avoids some necessary model descriptions.

My expertise is not in the region, but rather with geology in general (specifically, carbonate rocks), MODFLOW modelling and generalized model techniques and development. Therefore, I chose to focus on the model limitations, design, application and suitability and interpretation of results as other reviewers seemed to have had their say in the regional details.

The biggest hurdle in reading this paper was tracking down important information to adequately

evaluate their methods. Much of this (short geology description or more references, model design/structure) can be summarized in tables and figures, which does not add to the overall word count too much.

Please see attached document for review and questions to the authors about the model.

Review: Questions on model design.

1. Report a bit more details on the model design. I understand the model was developed independently outside the scope of this analysis (by the author), but important features such as spatial discretization, layers, whether there was an unconfined/convertible/confined layer for the top layer, boundary conditions and assumptions for certain features (i.e. lakes, drains, specified heads/fluxes), hydraulic properties assigned, among other important details help people get a proper handle on interpreting results and where errors may occur. There were clear alterations from the original DV3-SS model and identifying what changed and remained the same should be clearer. This is particularly problematic as when you go about changing boundary conditions the solutions can also change. These data can simply be summarized in a table and in some cases labeled in figures such as Figure 1.

2. I see all layers were simulated as confined layers in the model I downloaded (https://water.usgs.gov/GIS/metadata/usgswrd/XML/jackson2022_CEE.xml) I had assumed this was done simply to linearize the solutions and better handle inevitable drying of the top layer(s) due to the significant fluctuations in reported head (>10m).

However, I was surprised to find the starting heads of 0 would be embedded between both the 5th and the 6th layer within the model structure based on elevations of each of the layers. This means that wetting (and possibly drying) of cells occurs during convergence which can cause numerical instability/error and potential nonunique solutions (see any MODFLOW documentation). I also found incredibly isolated high heads in some parts of the model domain which were >7000ft (Figure S1). This generates seemingly unrealistic hydraulic gradients within the domain as well as possible unrealistic mass balance. This could be related to boundary condition issues which were changed between calibrated models and this one. How did the authors address the inherent numerical instability and error as well as potential nonunique solutions emerging from this model structure? Uncertainty analysis?

3. The no flow boundary indicates that the authors expect no contributing groundwater flow areas to Devils Hole outside the active model domain, deep or shallow. The authors suggest that the possibility for interbasin flow is negligible under modern conditions and less so under wetter periods. However, the authors seem to ignore that deeper sources could contribute to flow, particularly in fractured systems (Moore et al., 2009, Martinez-Santos et al., 2012, Gabrovsek and Dreybrodt, 2021). In fact, it appears the Pahrump valley region (to the south and not simulated) shares a deeper carbonate reservoir that crosses basin boundaries and could provide groundwater fluxes into the model domain and potentially subsequent upwelling under different water table elevations. If that is the case, water fluxes contributing to calculated head at Devils Hole would be ignored. Was there additional reasoning why interbasin flow was not considered? See Winograd and Doty, 1985.

Figure R1: Layer one with simulated heads for high recharge simulation (+9.5m or 244% recharge). Most cells are dry in the top layer in this scenario

4. How was the recharge simulated to generate the different scenarios? Was the model run a few hundred times with recharge being different each simulation and then the water level at Devils hole extracted? Or is it the equilibration period defined because they continuously run the model with a different recharge stressor every 1000 years? I am assuming some derivative of the former but would want that to be clearer as that has impact on the solutions. Particularly as the latter would cause some serious solution errors.
5. The authors mention that one of the limitations is the assumption of unchanged K in the model over 350k yrs, particularly under increased recharge. I agree this is a long time to go without changing aquifer properties in a tectonically active region, and especially for

carbonates which can have extensive porosity enhancement in shorter timescales which is enhanced by increased recharge (Ford and Williams, 2013).

However, I was confused by their assumption it is unquantifiable when a sensitivity/uncertainty analysis that includes K could help constrain some of the uncertainty in that assumption and add to the discussion. Was a sensitivity of K in the model performed? Perhaps this was this done in the original DV3-SS model and the authors could describe the results.

Review: Conceptual questions and figures

1. What exactly is the overall benefit of the study? If simulating past recharge cannot provide us useful information about what to expect from future, more arid climates over a broader scale, is there a reason other than academic curiosity that makes this useful beyond just a regional study?
2. The results in Figure 2 seem to indicate that the water table was never below current levels (0). However, in the results they report the water table did indeed fall to -1.6 m bcl and this is also shown in Figure 3. Did I miss something in the explanation of Figure 2?
3. Figure 2: The information about recharge and water table is redundant and the same information can be identified from Figures 3 & 4. More useful information on the figure would include the timing of the glacial and interglacial periods so the connection between water table and relationships to climate cycles can be made.
4. A value of 300% of current recharge sounds high but I also understand that number needs context. Can the authors provide what that recharge depth might be, on average or shown on a figure to give spatial variability to provide context on the 300% increase? Volumes over a large area can be misleading.

Summary

Overall, I agree with the previous reviews that the idea is novel, but the methods need a few more clarifications to be deemed reliable. A model that assumes 350k years where all boundaries, bulk calculated aquifer properties, and recharge variability remain static while not testing how changing them alters their solutions is a bit ambitious. Particularly as the intent of the model was to understand how climate impacts water levels and make decisions based on these results.

References:

Ford, D. and Williams, P.D., 2007. *Karst hydrogeology and geomorphology*. John Wiley & Sons.

Gabrovšek, F. and Dreybrodt, W., 2021. Early hypogenic carbonic acid speleogenesis in unconfined limestone aquifers by upwelling deep-seated waters with high CO₂ concentration: a modelling approach. *Hydrology and Earth System Sciences*, 25(5), pp.2895-2913.

Martínez-Santos, M., Ruíz-Romera, E., Martínez-López, M. and Antigüedad, I., 2012. Influence of upwelling on the shallow water chemistry in a small wetland riparian zone (Basque Country). *Applied geochemistry*, 27(4), pp.854-865.

Moore, P.J., Martin, J.B. and Sreaton, E.J., 2009. Geochemical and statistical evidence of recharge, mixing, and controls on spring discharge in an eogenetic karst aquifer. *Journal of Hydrology*, 376(3-4), pp.443-455.

Winograd, I.J. and Doty, G.C., 1980. *Paleohydrology of the southern Great Basin, with special reference to water table fluctuations beneath the Nevada Test Site during the late Pleistocene* (No. USGS-OFR-80-569). Geological Survey, Reston, VA (United States).

United States Department of the Interior

U.S. GEOLOGICAL SURVEY
Nevada Water Science Center
2730 N Deer Run Rd
Carson City, NV 89701

November 29, 2022

To: Joe Aslin
Senior Editor, Communications Earth & Environment

From: Tracie R. Jackson¹, Simon D. Steidle² et al.
¹ USGS Nevada Water Science Center, Boulder City, NV
² Institute of Geology, University of Innsbruck, Innsbruck, Austria

Subject: TECHNICAL REVIEW RESPONSE LETTER– Communications Earth & Environment Journal article titled “A 350,000-year history of groundwater recharge in the southern Great Basin, USA” by T.R. Jackson, S.D. Steidle, K.A. Wendt, Y. Dublyansky, R.L. Edwards, and C. Spötl

We thank the reviewers of this manuscript for their constructive comments. Text revisions to the original manuscript are in a “track-changes” copy of the manuscript, “DH2_WT-MODFLOW_v5.1_TrackedChange.docx”. Responses to comments from the three reviewers are provided below.

Reviewer #2 (Remarks to the Author):

I re-read the paper and rebuttal document from Jackson et al. and largely agree with their changes. I appreciate the thoughtful responses. Three minor points:

1) I still don't find figure 4's comparison to geologic data particularly satisfying, if the model was representing reality then a more quantitative comparison to the spatial data should be possible. That said, I understand that this is not the point of the paper.

Answer: We agree that comparing modeled paleo-discharge to geologic data is not an overly satisfying way to verify the model, however, we do not know of a better way. The only "ground truth" that can be compared to the model are spatially mapped paleo-discharge locations, which has been done in this work. Discharge rates from these paleo-discharge areas are not known and have not been estimated in previous work. Therefore, we cannot compare model-estimated values to measured discharge rates. (no further changes made to the manuscript)

2) Further the inability to place some level of uncertainty on the estimates based on model assumptions is somewhat unsatisfying, but again I understand how complex it is to run such models, and thus such a sensitivity test can be left for another paper.

Answer: We did a partial uncertainty analysis to address this comment and comment #3. One of the primary uncertainties when estimating wetter and drier climates is the distribution of recharge. We ran a scenario with a uniform increase in recharge (as suggested in comment #3) to compare to the scaled recharge scenario. Results of the uncertainty analysis with respect to the recharge distribution are presented in the paper and demonstrate that the model results are robust. General agreement between simulated and mapped paleo-discharge areas indicates that scaled- and uniform-recharge distributions can reasonably simulate glacial conditions, whereas scaled recharge better simulates interglacial conditions. The relation between Devils Hole water-level changes and percent change in recharge remains mostly unchanged.

3) Finally, the rebuttal to why a recharge multiple is convincing, but what about a uniform increase in recharge (in units of say mm/yr) across the domain? Would that be more realistic? If you think about Clausius–Clapeyron scaling of the water cycle (likely that is too sensitive) delivery of water to a region under warmer or colder periods should scale by amount not a multiple right?

Answer: We believe that using a recharge multiplier best represents the spatial distribution of paleo-recharge. The same mountains and valleys existed within the last 350,000 years in the study area, and there is no reason to believe that the recharge mechanisms have changed much within this period. At present, the mountains accumulate small snowpacks, which are the primary contributor to groundwater recharge. We believe that during wetter and colder periods, these same mountains would have held very large snowpacks that would have contributed the bulk of recharge to the groundwater system. Therefore, using a multiplier seems most reasonable. However, since the distribution of recharge in different climates is not known, we also simulated additional recharge by uniformly adding it to or subtracting it from the existing recharge distribution. By using the "uniform" approach, the recharge contribution from the mountains is less significant relative to the valleys during wet periods. Although the "uniform" approach does not fundamentally change the conclusions in the paper, the greater emphasis on recharge in the valley floors results in larger paleo-discharge footprints in the modeled results (see new Figures 4 and 5). Note that these new figures show paleo-discharge areas simulated using the scaled- and uniform-recharge approaches for peak glacial and interglacial scenarios. The abstract, methods, results, discussion, and conclusions have been revised to incorporate the uniform-recharge discussion.

I make a few final (and very minor suggestions below) related to changes and look forward to future group from this group on the (transient) modeling of this system.

Minor comments/changes:

121-123: You may wish to note though that some of the work since Wendt 2018 (Wendt et al., 2020 and Li et al., 2020) show systematic changes in systems such as the ($^{234}\text{U}/^{238}\text{U}$)initial (also denoted $d^{234}\text{U}$) that provide a more 'continuous' history of recharge, albeit with some delay and geochemical complications

Answer: We added Wendt et al. (2020) and the following sentence to the introduction: "At Devils Hole, Wendt et al. (2020) qualitatively related the Devils Hole paleo-water-table record to recharge, but the study is not suitable for quantifying basin-scale recharge volumes." It is an interesting thought to connect this study with recharge (though certainly not straight forward) and we are thankful for the suggestion. Li et al. (2020) has a different focus and we find it less relevant in this context.

193: change to climate models or global climate models

Answer: Changed "atmospheric models" to "climate models"

Map figures: these are somewhat grainy and the text pretty small for this size of a figure. Consider adjusting for publication.

Answer: Thank you for bringing this to our attention. We will certainly check and verify the quality and readability for the proofs.

Reviewer #3 (Remarks to the Author):

Overall this remains an important paper. I do feel there were some lost opportunities in the revision to handle some of the reviewers' comments.

Some points seemed to mainly be argued in the response to reviewers rather than modifying the manuscript to account for the suggestions. One example is the reviewer question beginning "117-131: But glacial-interglacial cycles show abrupt changes on <10 kyrs timescales..." How did the authors modify the manuscript to account for this important observation?

Answer: This review comment was related to the justification for using a steady-state simulation to estimate recharge during past glacial and interglacial periods. The use of a steady-state model to simulate a changing climate is now acknowledged in the manuscript and is fully addressed in a Supplementary Information accompanying the manuscript. The lengthy explanation that justifies the steady-state assumption can be found in Supplementary Information section "Justification for Steady-State Model Assumption".

Another is the evaluation of data relating to the question of Mid Holocene aridity. Was the manuscript modified to account for this important point? If the DH folio do not record the enhance mid-Holocene aridity then one has to question how well they were able to account for other millennial-scale aridity intervals. What I would like to see is how the manuscript was modified to account for this point, but I did not see where in the manuscript this was handled. Nor how the manuscript was modified to address the issue of interbasin flow from Railroad Valley and other locations. At the least it is necessary to state in the manuscript that a lack of interbasin flow into the modeled basins from further north is an assumption.

Answer: This paper relies on the data published by Wendt et al. 2018. The temporal resolution and quality of the record with regard to millennial-scale events is thus the same as in Wendt et al., 2018. For a full discussion on calcite sampling limitations and the potential for "missing" arid events in the discontinuous record, see Wendt et al. 2018. It is beyond the scope of this paper to account for or even discuss water table proxy data in the form of calcite deposits. Regarding why interbasin flow is negligible, the following paragraph was added to the last paragraph of the section "Study Area Description":

"Expanding the study area to include adjacent bounding basins would have no effect on the water levels in Devils Hole. For example, south of Devils Hole, interbasin flow from Pahrump Valley is negligible under modern conditions because geologic and hydrologic evidence indicate that a low-permeability hydraulic barrier precludes groundwater movement northward from Pahrump Valley to Devils Hole (Fig. 1; Winograd and Thordarson, 1975; Fenelon et al., 2016). Therefore, interbasin flow does not occur near Devils Hole. Devils Hole water levels also will be unaffected by potential, but unlikely, interbasin flows into the study area from basins distant from Devils Hole, such as Railroad Valley (Fenelon et al., 2016; Halford and Jackson, 2020; Jackson et al., 2021). During wetter periods, groundwater-flow paths shorten between recharge and discharge areas, and more discharge areas form. Therefore, the contributing area to Devils Hole will be smaller. During drier periods, significant interbasin flows are unlikely across basin boundaries because modern discharge areas will either decrease in areal extent, remain the same size but have decreased discharge rates, or disappear due to lowered water levels. Therefore, inclusion of the AMGB and AFFCRGB is adequate for estimating recharge volumes under different climate scenarios. See Supplementary Information section "Study Area" for a more detailed explanation that justifies the study area extent."

Reviewer #4 (Remarks to the Author):

The authors take a novel approach to simulating potential recharge values that could explain estimated paleo water table levels throughout a 350k yr span. The authors converted an existing groundwater model for the AMGB to a superposition model to analyze changes from current, baseline values. The model design was such that relative changes in water level under different recharge values ranging from -24-300% from current estimated recharge across the basin were analyzed. The method was to develop a relationship between recharge and expected water level and determine what recharge would generate observed water levels at a single point in Devils Hole where a paleo-water table record existed.

The model is unique, and I think the study and results are intriguing. The article goes out of its way to avoid too much modelling jargon. However, as I address later, it also avoids some necessary model descriptions.

My expertise is not in the region, but rather with geology in general (specifically, carbonate rocks), MODFLOW modelling and generalized model techniques and development. Therefore, I chose to focus on the model limitations, design, application and suitability and interpretation of results as other reviewers seemed to have had their say in the regional details.

The biggest hurdle in reading this paper was tracking down important information to adequately evaluate their methods. Much of this (short geology description or more references, model design/structure) can be summarized in tables and figures, which does not add to the overall word count too much.

Answer: To adequately address this comment, a Supplementary Information section was written and included with the manuscript. The Supplementary Information includes discussion of the geology, hydrogeology, and modern predevelopment flow setting to understand the groundwater-flow conceptualization. A summary of the model design, boundary conditions, and calibration of the DV3-SS model also is provided. Modifications to the DV3-SS model are documented. Tables, figures, and text are used to provide clarity for readers interested in the modelling mechanics. We agree that providing a concise description of model methods with the paper will alleviate the need for readers to read background material. However, we chose to document the published DV3-SS model in Supplementary Information because adding it to the paper would make the paper prohibitively long and cause the reader to lose focus on the main study objective.

Please see attached document for review and questions to the authors about the model: *[.pdf pasted below]*

Review: Questions on model design.

1. Report a bit more details on the model design. I understand the model was developed independently outside the scope of this analysis (by the author), but important features such as spatial discretization, layers, whether there was an unconfined/convertible/confined layer for the top layer, boundary conditions and assumptions for certain features (i.e. lakes, drains, specified heads/fluxes), hydraulic properties assigned, among other important details help people get a proper handle on interpreting results and where errors may occur. There were clear alterations from the original DV3-SS model and identifying what changed and remained the same should be clearer. This is particularly problematic as when you go about changing boundary conditions the solutions can also change. These data can simply be summarized in a table and in some cases labeled in figures such as Figure 1.

Answer: *To adequately address this comment, a Supplementary Information section was written and included with the manuscript. The Supplementary Information includes discussion of the geology, hydrogeology, and modern predevelopment flow setting to understand the groundwater-flow conceptualization. A summary of the model design, boundary conditions, and calibration of the DV3-SS model is provided as well as modifications to the DV3-SS model. Documentation of the published DV3-SS model in this paper would make the paper prohibitively long and cause the reader to lose focus on the main study objective. A table is not sufficient to fully describe the original DV3-SS model in the paper and adding this information to figure 1 would make the figure difficult to read. Therefore, details of the original model design and modifications are included in a new Supplementary Information section.*

2. I see all layers were simulated as confined layers in the model I downloaded (https://water.usgs.gov/GIS/metadata/usgswrd/XML/jackson2022_CEE.xml) I had assumed this was done simply to linearize the solutions and better handle inevitable drying of the top layer(s) due to the significant fluctuations in reported head (>10m). However, I was surprised to find the starting heads of 0 would be embedded between both the 5th and the 6th layer within the model structure based on elevations of each of the layers. This means that wetting (and possibly drying) of cells occurs during convergence which can cause numerical instability/error and potential nonunique solutions (see any MODFLOW documentation). I also found incredibly isolated high heads in some parts of the model domain which were >7000ft (Figure S1). This generates seemingly unrealistic hydraulic gradients within the domain as well as possible unrealistic mass balance. This could be related to boundary condition issues which were changed between calibrated models and this one. How did the authors address the inherent numerical instability and error as well as potential nonunique solutions emerging from this model structure? Uncertainty analysis?

Answer: The simulated heads are correct, and the use of confined layers for this model exercise is consistent with best modeling practices (Anderson and Woessner, 2002). Notice in the MODFLOW LIST file that no convergence errors or numerical instability has occurred during model convergence. Furthermore, our mass balance error is less than 1 percent, more than acceptable. As stated in the paper,

“The modified model was developed by converting the original DV3-SS model into a superposition model. The model has initial heads of 0 m, which are conceptualized as no water-level change from modern conditions. Simulated water-level changes are relative to modern heads, where the modern (predevelopment) head distribution was obtained from the calibrated DV3-SS model (Halford and Jackson, 2020).”

We are aware that there are a few isolated high head values in Bare Mountain. This is expected. Recharge volumes are limited where hydraulic conductivity of bedrock is less than infiltration rates. Halford and Jackson (2020) calibrated recharge rates by accounting for the effects of hydraulic conductivity on simulated heads. During calibration, recharge rates could not exceed hydraulic conductivity of saturated rocks, which results in excess infiltration being displaced laterally downgradient in Belted, Groom, Eleana, and Greenwater Ranges and areas of the Spring Mountains. For this paper, recharge was added to model cells, without considering hydraulic conductivities in low-permeability highland areas. These spurious heads in a few model cells have no effect on understanding the relation between Devils Hole water levels and recharge volumes.

When interpreting model results, the paper states:

“A volumetric modern-recharge rate of 2.59×10^7 m³/yr (Halford and Jackson, 2020) is the reference recharge (0%) and corresponds to the modern level (0 m r.m.w.t.) for the purpose of estimating percent change in recharge relative to modern.”

To demonstrate how to interpret model results, a simulated head distribution from the glacial (+9.5 m) scenario is shown in Figure 1 below. The simulated head distribution is not an absolute water-level altitude map, but a map of head change from modern conditions. If heads are zero through all model layers (layers 1–6), then this means that the heads have not changed from modern heads. Low-permeability rocks occur in the highland areas of the Belted, Groom, Eleana, and Sheep Ranges and in the Black and Spring Mountains (Figure 2). In these low-permeability highland areas, recharge rates in the glacial (+9.5 m) scenario exceed the hydraulic conductivity of saturated rocks, which is expected to cause large head changes. Therefore, the head-change distribution in Figure 1 follows expectations of steep hydraulic gradients in highland recharge areas with low-permeability rocks and gentle hydraulic gradients in more permeable areas. A discussion is added to the Supplementary Information titled “Interpreting Results from Modified DV3-SS Model” to help readers understand the concept of superposition.

Figure 1. Glacial scenario simulating a 9.5-m rise in Devils Hole, relative to modern heads at 0 m.

Figure 2. DV3 hydrogeologic framework in DV3-SS model.

3. The no flow boundary indicates that the authors expect no contributing groundwater flow areas to Devils Hole outside the active model domain, deep or shallow. The authors suggest that the possibility for interbasin flow is negligible under modern conditions and less so under wetter periods. However, the authors seem to ignore that deeper sources could contribute to flow, particularly in fractured systems (Moore et al., 2009, Martinez-Santos et al., 2012, Gabrovsek and Dreybrodt, 2021). In fact, it appears the Pahrump valley region (to the south and not simulated) shares a deeper carbonate reservoir that crosses basin boundaries and could provide groundwater fluxes into the model domain and potentially subsequent upwelling under different water table elevations. If that is the case, water fluxes contributing to calculated head at Devils Hole would be ignored. Was there additional reasoning why interbasin flow was not considered? See Winograd and Doty, 1985.

Answer: Expanding the model domain to include areas such as Pahrump Valley would have no effect on the water levels in Devils Hole either during modern conditions or past climatic conditions. The assumption that more than negligible amounts of interbasin flow occurs from Pahrump Valley to Ash Meadows basin (and Devils Hole) is incorrect based on geologic and hydrologic evidence. Below is a summary of this evidence, from the landmark paper by Winograd and Thordarson (1975).

1. First, a lower clastic aquitard crops out along the southern border of the Ash Meadows basin boundary between Pahrump and Ash Meadows basin, forming a hydraulic barrier to groundwater flow. The distribution of these low-permeability clastic rocks is controlled by the Montgomery thrust fault. The upper plate of the thrust fault consists of late Precambrian clastic rocks that dip westward and northwestward, whereas the lower plate consists of Devonian to Mississippian carbonate rocks. Thus, this thrust fault isolates groundwater in the carbonate aquifer in Pahrump Valley from carbonate rocks in Ash Meadows basin (see figure 3 below from Sweetkind et al., 2001). Blue colors are carbonate rocks, brown colors are clastic rocks).

Figure 3. Geologic section showing the Montgomery thrust, which isolates carbonate rocks in Pahrump Valley basin from carbonate rocks in Ash Meadows basin.

2. Second, the difference in groundwater levels between northwestern Pahrump Valley and Devils Hole is about 200 ft. The large difference in heads between the two areas strongly suggests groundwater damming and a lack of hydraulic connection. Estimated transmissivities of clastic rocks from slug tests are on the order of $0.00001 \text{ ft}^2/\text{d}$ and estimated interbasin flow, based on Darcy's Law, is about 1 gal/min, or 0.0076% of the Ash Meadows basin flow budget.

3. *Third, depth to water in northwestern Pahrump Valley (near the Ash Meadows basin boundary) suggest that this valley is separated from Ash Meadows by a relatively impermeable barrier. The water table beneath northwestern Pahrump Valley is shallow (within a few feet of land surface. The water table in the basin-fill aquifer in northwestern Pahrump Valley thus stands well above levels in the same aquifer to the south-southeast. Saturation of the valley fill nearly to the surface beneath the playa in northwestern Pahrump Valley suggests that groundwater in this area is ponded by an impermeable boundary, namely, the lower clastic aquitard. Such ponding does not preclude underflow of small magnitude (~1 gal/min).*
4. *Fourth, the chemical quality of groundwater in Pahrump Valleys precludes groundwater movement of significant quantities into Ash Meadows basin. See Winograd and Thordarson (1975, p. C108–C109) for details.*

The current study includes Ash Meadows and AFFCR groundwater basins because Halford and Jackson (2020) concluded that these basins are coupled and are better considered as a single groundwater basin. The PDVS groundwater basin, which includes Pahrump Valley, was excluded from the current study because the basin is hydraulically isolated from the Ash Meadows and AFFCR groundwater basins based on the reasoning described above. The PMOV groundwater basin was excluded from the current study because the basin largely is hydraulically isolated from the neighboring Ash Meadows and AFFCR groundwater basins. Basin budget analyses (Fenelon et al., 2016; Jackson et al., 2021) and groundwater modeling of PMOV basin boundaries (Fenelon et al., 2016; Halford and Jackson, 2020) indicate that negligible interbasin flow occurs between the PMOV–Ash Meadows boundary (less than 0.1%), and insignificant interbasin flow occurs between the PMOV–AFFCR boundary (less than 2%).

Interbasin flows from other bounding basins also were ruled out for geologic and hydrologic reasons. See the most recent interpretations from Fenelon et al. (2016) and Halford and Jackson (2020). Older reports published before the more recent work of Fenelon et al. (2016) and Halford and Jackson (2020) are outdated because they are based on low-accuracy reconnaissance data, limited datasets that do not fully characterize the system, or unfounded conceptualizations that have since been debunked, such as significant volumes of interbasin flows that occur deep within the Earth. The groundwater-flow conceptualization has significantly changed within the last decade based on recent work. We encourage the reviewer to read the Introduction section of Halford and Jackson (2020), which provides a history of how the science has changed, to date, in the Death Valley Regional Flow System. In summary, interpretation of high-quality, decadal water-level records from a large water-level monitoring network and high-accuracy, micrometeorological estimates of groundwater discharge from discharge areas have changed flow-path interpretations. Large-scale multiple-well aquifer tests done on the Nevada National Security Site also have refined hydraulic-property estimates. In section “Study Area Description”, a paragraph was added:

“Expanding the study area to include adjacent bounding basins would have no effect on the water levels in Devils Hole. For example, south of Devils Hole, interbasin flow from Pahrump Valley is negligible under modern conditions because geologic and hydrologic evidence indicate that a low-permeability hydraulic barrier precludes groundwater movement northward from Pahrump Valley to Devils Hole (Fig. 1; Winograd and Thordarson, 1975; Fenelon et al., 2016). Therefore, interbasin flow does not occur near Devils Hole. Devils Hole water levels also will be unaffected by potential, but unlikely, interbasin flows into the study area from basins distant from Devils Hole, such as Railroad Valley (Fenelon et al., 2016; Halford and Jackson, 2020; Jackson et al., 2021). During wetter periods, groundwater-flow paths shorten between recharge and discharge areas, and more discharge areas form. Therefore, the contributing area to Devils Hole will be smaller. During drier periods, significant interbasin flows are unlikely across basin boundaries because modern discharge areas will either decrease in areal extent, remain the same size but have decreased discharge rates, or disappear due to lowered water levels. Therefore, inclusion of the AMGB and AFFCRGB is adequate for estimating recharge volumes under different climate scenarios. See Supplementary Information section “Study Area” for a more detailed explanation that justifies the study area extent.”

Fenelon, J.M., Halford, K.J., and Moreo, M.T., 2016, Delineation of the Pahute Mesa–Oasis Valley groundwater basin, Nevada (ver. 1.1, May 2016): U.S. Geological Survey Scientific Investigations Report 2015–5175, 40 p., <http://dx.doi.org/10.3133/sir20155175>.

Halford, K.J., and Jackson, T.R., 2020, Groundwater characterization and effects of pumping in the Death Valley regional groundwater flow system, Nevada and California, with special reference to Devils Hole: U.S. Geological Survey Professional Paper 1863, 178 p., <https://doi.org/10.3133/pp1863>.

Jackson, T.R., Fenelon, J.M., and Paylor, R.L., 2021, Groundwater flow conceptualization of the Pahute Mesa–Oasis Valley Groundwater Basin, Nevada—A synthesis of geologic, hydrologic, hydraulic-property, and tritium data: U.S. Geological Survey Scientific Investigations Report 2020–5134, 100 p., <https://doi.org/10.3133/sir20205134>.

Winograd, I.J., and Thordarson, W., 1975, Hydrogeologic and hydrochemical framework, south-central Great Basin, Nevada-California, with special reference to the Nevada Test Site: U.S. Geological Survey Professional Paper 712-C, 126 p.

4. How was the recharge simulated to generate the different scenarios? Was the model run a few hundred times with recharge being different each simulation and then the water level at Devils hole extracted? Or is it the equilibration period defined because they continuously run the model with a different recharge stressor every 1000 years? I am assuming some derivative of the former but would want that to be clearer as that has impact on the solutions. Particularly as the latter would cause some serious solution errors.

Answer: *Yes, the model was run a few hundred times with recharge being different each simulation (and then the water level at Devils hole was extracted). A sentence was added to the Methods to state:*

“In summary, the calibrated, modern recharge distribution was multiplied by a scaling factor, then the model was run, the simulated head at Devils Hole was extracted, and the process repeated to obtain a relation between recharge and Devils Hole water levels.”

A uniform-recharge scenario was added to the revised paper, based on suggestions from Reviewer #2. For the uniform-recharge scenario, a series of recharge arrays were created by adding a uniform recharge amount in increments of 0.0001 m from -0.0018 to 0.0066 m to the calibrated, modern recharge distribution. For clarity of this comment, the following text was added:

“For each model run, one recharge array was called by the MODFLOW recharge package, and the simulated head at Devils Hole was extracted. The process was repeated, with one model run for each recharge array, to obtain a relation between recharge and Devils Hole water levels.”

5. The authors mention that one of the limitations is the assumption of unchanged K in the model over 350k yrs, particularly under increased recharge. I agree this is a long time to go without changing aquifer properties in a tectonically active region, and especially for carbonates which can have extensive porosity enhancement in shorter timescales which is enhanced by increased recharge (Ford and Williams, 2013). However, I was confused by their assumption it is unquantifiable when a sensitivity/uncertainty analysis that includes K could help constrain some of the uncertainty in that assumption and add to the discussion. Was a sensitivity of K in the model performed? Perhaps this was this done in the original DV3-SS model and the authors could describe the results.

Answer: *Thank you for your comment. The section "Model Assumptions and Uncertainties" was changed to "Model Assumptions" and the entire section rewritten. The following arguments are used to justify the model assumptions:*

"This work assumes that changes in rock transmissivity over the past 350,000-years were minimal and did not significantly affect the Devils Hole paleo-water-table record. This work also assumes that the water-table record was not significantly affected by tectonic deformation that caused local changes to the land-surface altitude or volumetric strain in the aquifer. Carbonate-rock dissolution or fracturing and faulting due to tectonic activity could alter transmissivity (Winograd and Doty, 1980). Incremental changes in transmissivity, land-surface adjustments, or volumetric strain over the past 350,000 years are expected to impose a long-term downward or upward trend in the Devils Hole water-level record. For example, Robertson et al. (2007) estimated that the rate of water-level decline in Devils Hole due to volumetric strain could be as high as 0.02 cm/yr. If this rate were sustained for 350,000 years, the water level in Devils Hole would have declined 70 m. A water-level change of this magnitude is not observed (Fig. 2A). The water-level record is dominated by large oscillations that correspond to wet and dry climatic conditions, whereas the long-term trend is relatively flat. Any potential long-term rise or decline that is masked by these large oscillations would have to be small (<2 m over 350,000 years). Therefore, the assumption is reasonable that changes to transmissivity, land-surface altitude, and volumetric strain over the past 350,000 years were minimal relative to changes in the Devils Hole water-table record."

A formal sensitivity analysis has not been done and is beyond the scope of this work. To do a proper analysis, K estimates would need to be varied in all fractured rocks (volcanic and carbonate). Winograd and Doty (1980, table 2) permit a transmissivity change resulting from dissolution of carbonate rock during the pluvial period of up to 50% of modern estimates for their uncertainty analysis, but state they do not believe such a change actually occurred. Scenarios would need to be run that vary K estimates within this bounding transmissivity range. Note that faulting may have caused K to increase or decrease with time and investigation of structures also may need to be considered. Structural effects on transmissivity are heterogeneous, such that spatially varying changes in transmissivity also would need to be considered. A proper uncertainty analysis is a scope of work in itself and is beyond the scope of the current paper.

Review: Conceptual questions and figures

1. What exactly is the overall benefit of the study? If simulating past recharge cannot provide us useful information about what to expect from future, more arid climates over a broader scale, is there a reason other than academic curiosity that makes this useful beyond just a regional study?

Answer: This study indeed aims at improving the (local) understanding of a more arid climate in the future (e.g. the interglacial sensitivity mentioned in the abstract). Throughout the review process we focused more on the most reliable aspects, and we try to not overstate the results of the model.

Reconstructing last glacial maximum (LGM) conditions in the Great Basin has been a topic of scientific interest for over a century. The most recent iteration of global climate models (PMIP3) show a general lack of agreement in LGM precipitation anomalies in the Great Basin, underlining the need for quantitative estimates on paleo-hydrological conditions. This study contributes to this ongoing effort. Additionally, this study can serve as a blueprint for other regions. This study is certainly not the end of the line.

2. The results in Figure 2 seem to indicate that the water table was never below current levels (0). However, in the results they report the water table did indeed fall to -1.6 m bcl and this is also shown in Figure 3. Did I miss something in the explanation of Figure 2?

Answer: Figure 2 has been completely redrafted for clarity to show censoring in the paleo-water-table record for clarity.

3. Figure 2: The information about recharge and water table is redundant and the same information can be identified from Figures 3 & 4. More useful information on the figure would include the timing of the glacial and interglacial periods so the connection between water table and relationships to climate cycles can be made.

Answer: Glacial and interglacial periods have been added to the redrafted figure 2.

4. A value of 300% of current recharge sounds high but I also understand that number needs context. Can the authors provide what that recharge depth might be, on average or shown on a figure to give spatial variability to provide context on the 300% increase? Volumes over a large area can be misleading.

Answer: Figures S7–S10 have been added to the Supplementary Information section, which show recharge-rate distributions for peak glacial and interglacial conditions using the scaled- and uniform-recharge approaches.

Summary

Overall, I agree with the previous reviews that the idea is novel, but the methods need a few more clarifications to be deemed reliable. A model that assumes 350k years where all boundaries, bulk calculated aquifer properties, and recharge variability remain static while not testing how changing them alters their solutions is a bit ambitious. Particularly as the intent of the model was to understand how climate impacts water levels and make decisions based on these results.

Answer: The paper presents a novel approach for estimating recharge during different climate regimes. A partial-uncertainty analysis has been added to the revised manuscript to constrain uncertainty associated with recharge. Including additional uncertainty and sensitivity analyses associated with K is beyond the scope (see response to comment #5 above).

References:

Ford, D. and Williams, P.D., 2007. *Karst hydrogeology and geomorphology*. John Wiley & Sons.

Gabrovšek, F. and Dreybrodt, W., 2021. Early hypogenic carbonic acid speleogenesis in unconfined limestone aquifers by upwelling deep-seated waters with high CO₂ concentration: a modelling approach. *Hydrology and Earth System Sciences*, 25(5), pp.2895-2913.

Martínez-Santos, M., Ruíz-Romera, E., Martínez-López, M. and Antigüedad, I., 2012. Influence of upwelling on the shallow water chemistry in a small wetland riparian zone (Basque Country). *Applied geochemistry*, 27(4), pp.854-865.

Moore, P.J., Martin, J.B. and Sreaton, E.J., 2009. Geochemical and statistical evidence of recharge, mixing, and controls on spring discharge in an eogenetic karst aquifer. *Journal of Hydrology*, 376(3-4), pp.443-455.

Sweetkind, D.S., Dickerson, R.P., Blakely, R.J., and Denning, P.D., 2001, Interpretive geologic cross sections for the Death Valley regional flow system and surrounding areas, Nevada and California: U.S. Geological Survey Miscellaneous Field Studies Map MF-2370, 38 p, 3 sheets.

Winograd, I.J. and Doty, G.C., 1980. *Paleohydrology of the southern Great Basin, with special reference to water table fluctuations beneath the Nevada Test Site during the late Pleistocene* (No. USGS-OFR-80-569). Geological Survey, Reston, VA (United States).

31st Jan 23

Dear Mr Steidle,

Your manuscript titled "A 350,000-year history of groundwater recharge in the southern Great Basin, USA" has now been seen by our reviewers, whose comments appear below. In light of their advice I am delighted to say that we are happy, in principle, to publish a suitably revised version in Communications Earth & Environment under the open access CC BY license (Creative Commons Attribution v4.0 International License).

We therefore invite you to revise your paper one last time to address the remaining concerns of our reviewers. In particular, we concur with Reviewer #3 that your findings should be placed in the context of the possibility that water tables may have been lower during Mid-Holocene peak arid intervals. Please ensure this is discussed in the manuscript and accounted for in the conclusions.

At the same time we ask that you edit your manuscript to comply with our format requirements and to maximise the accessibility and therefore the impact of your work.

EDITORIAL REQUESTS:

*****Please take care to match our formatting and policy requirements. We will check revised manuscript and return manuscripts that do not comply. Such requests will lead to delays. *****

SUBMISSION INFORMATION:

OPEN ACCESS:

Communications Earth & Environment is a fully open access journal. Articles are made freely accessible on publication under a [CC BY license](http://creativecommons.org/licenses/by/4.0) (Creative Commons Attribution 4.0 International License). This license allows maximum dissemination and re-use of open access materials and is preferred by many research funding bodies.

For further information about article processing charges, open access funding, and advice and

support from Nature Research, please visit <https://www.nature.com/commsenv/article-processing-charges>

At acceptance, you will be provided with instructions for completing this CC BY license on behalf of all authors. This grants us the necessary permissions to publish your paper. Additionally, you will be asked to declare that all required third party permissions have been obtained, and to provide billing information in order to pay the article-processing charge (APC).

[link redacted]

Best regards,

Joe Aslin

Senior Editor,
Communications Earth & Environment
<https://www.nature.com/commsenv/>
Twitter: @CommsEarth

REVIEWERS' COMMENTS:

Reviewer #2 (Remarks to the Author):

I am satisfied with the authors responses to my (minimal) additional comments, thank you for the thoughtful replies and changes that were made.

Reviewer #3 (Remarks to the Author):

The revised version of the manuscript continues to provide strong evidence for the model-based estimates of paleo-recharge in the Ash Meadows Groundwater Basin. Overall I remain positive about the contributions of the manuscript. This first-order analysis is an important contribution to the literature because it quantifies recharge amounts to past groundwater levels. On the first order, it can be used to estimate future water table changes associated with aridification as projected from climate model. The modeling data are only as good as the proxy water table levels as constrained by the calcite deposits (folia, mammillary) from Devils Hole. In my original review, I suggested the authors consider the possibility that water tables may have been even lower during peak arid intervals like the Middle Holocene, but this suggestion did not appear to make it into any revision, resulting in a missed opportunity to provide a broader context to potential future climate change.

On this point, the authors state in their rebuttal: “This paper relies on the data published by Wendt et al. 2018. The temporal resolution and quality of the record with regard to millennial-scale events is thus the same as in Wendt et al., 2018. For a full discussion on calcite sampling limitations and the potential for “missing” arid events in the discontinuous record, see Wendt et al. 2018. It is beyond the scope of this paper to account for or even discuss water table proxy data in the form of calcite deposits.”

On the contrary. It is fully within the scope of the current paper and indeed the peer review process to address the point in the current manuscript. At the least, I would suggest the following revision to the manuscript around line 142-143 of the track changes version: “For example, the lowest core sampled, at -1.6 m below the water table does not contain folia. However, water tables at or lower than -1.6 m below the water table cannot be ruled out, due to an absence of cores below this level or the possibility of a brief water table lowering that did not result in folia growth. Such a possibility for transient water tables at or below -1.6 m is supported by widespread evidence of aridity in the Great Basin that was greater than modern during the Middle Holocene.” Such contextual information could be added to the conclusion of the paper as well.

Reviewer #4 (Remarks to the Author):

I have reviewed the responses by the authors and am pleased with their efforts to address all reviewers concerns. I still have some concerns about the model structure and runs, however the answers they provided are adequate and address the more pressing issues that were brought up.

I am recommending the manuscript for publication without any further problems to address.

REVIEWERS' COMMENTS:

Reviewer #2 (Remarks to the Author):

Comment: I am satisfied with the authors responses to my (minimal) additional comments, thank you for the thoughtful replies and changes that were made.

Response: Thank you for your review.

Reviewer #3 (Remarks to the Author):

The revised version of the manuscript continues to provide strong evidence for the model-based estimates of paleo-recharge in the Ash Meadows Groundwater Basin. Overall I remain positive about the contributions of the manuscript. This first-order analysis is an important contribution to the literature because it quantifies recharge amounts to past groundwater levels. On the first order, it can be used to estimate future water table changes associated with aridification as projected from climate model. The modeling data are only as good as the proxy water table levels as constrained by the calcite deposits (folia, mammillary) from Devils Hole. In my original review, I suggested the authors consider the possibility that water tables may have been even lower during peak arid intervals like the Middle Holocene, but this suggestion did not appear to make it into any revision, resulting in a missed opportunity to provide a broader context to potential future climate change.

On this point, the authors state in their rebuttal: “This paper relies on the data published by Wendt et al. 2018. The temporal resolution and quality of the record with regard to millennial-scale events is thus the same as in Wendt et al., 2018. For a full discussion on calcite sampling limitations and the potential for “missing” arid events in the discontinuous record, see Wendt et al. 2018. It is beyond the scope of this paper to account for or even discuss water table proxy data in the form of calcite deposits.”

On the contrary. It is fully within the scope of the current paper and indeed the peer review process to address the point in the current manuscript. At the least, I would suggest the following revision to the manuscript around line 142-143 of the track changes version: “For example, the lowest core sampled, at -1.6 m below the water table does not contain folia. However, water tables at or lower than -1.6 m below the water table cannot be ruled out, due to an absence of cores below this level or the possibility of a brief water table lowering that did not result in folia growth. Such a possibility for transient water tables at or below -1.6 m is supported by widespread evidence of aridity in the Great Basin that was greater than modern during the Middle Holocene.” Such contextual information could be added to the conclusion of the paper as well.

Response: We agree with reviewer#3, that there is no proof that the water table never fell below -1.6 m. We have incorporated a statement about this in the conclusion:

*“The lower limit of the paleo water table was **most likely** above -1.6 m r.m.w.t. during peak interglacial conditions, which relates to a decrease in estimated recharge of no more than 17%, relative to modern, using the scaled-recharge assumption. A **water table lower than -1.6 m r.m.w.t. cannot be ruled out but is considered unlikely.**”*

Furthermore, we have updated the subsection, ‘Limitations of the paleo-water-table record’, which now incorporates additional arguments why we *interpret* -1.6 m as a lower limit while making clear, that there are scenarios which could have led to a water table even lower than -1.6 m. We also provide context by mentioning the Mid-Holocene aridity in the region. The section now reads as follows:

“The 350,000-year Devils Hole water-table record was constructed by Szabo et al. (1994)¹⁰ and Wendt et al. (2018)¹¹ using calcite cores that were sampled from cave walls at discrete elevations (Fig. 2a). Due to the nature of discrete sampling, the compiled record does not capture the precise elevation of paleo-water-table fluctuations. For example, the two lowest cores were sampled at +0 m and -1.6 m relative to the modern water table. The lowest core does not contain folia, nor are folia observed on the exposed cave walls below -1.6 m. Therefore, the paleo-water-table is assumed to have never dropped below -1.6 m. Yet it is possible that the paleo-water-table fluctuated between modern-day levels and -1.6 m during periods of greater aridity. For example, evidence suggests that the Great Basin was drier during the Middle Holocene relative to today²¹. This period is poorly resolved in the Devils Hole paleo-water-table record (~3kyr sampling resolution). Any potential folia deposits between +0 m and -1.6 m during the Middle Holocene cannot be discerned in the current record. Furthermore, rapid fluctuations of the paleo-water-table could result in microscopically thin or potential absence of folia deposits at sampled elevations¹¹. We acknowledge this lack of complete spatial and temporal continuity, and instead use the Devils Hole paleo-water-table record to investigate recharge conditions under a range of climate states, rather than individual events, during the last 350,000 years.”

Reviewer #4 (Remarks to the Author):

Comment: I have reviewed the responses by the authors and am pleased with their efforts to address all reviewers concerns. I still have some concerns about the model structure and runs, however the answers they provided are adequate and address the more pressing issues that were brought up.

I am recommending the manuscript for publication without any further problems to address.

Response: Thank you for your review.